# Southern Ocean Ice Prediction System version 1.0 (SOIPS v1.0): description of the system and evaluation of synoptic scale sea ice forecasts

Fu Zhao[1], Xi Liang[1], Zhongxiang Tian[1], Ming Li[1], Na Liu[1], Chengyan Liu[2]

[1]Key Laboratory of Marine Hazards Forecasting, National Marine Environmental Forecasting Center, Ministry of Natural Resources, Beijing, China.
[2]Southern Marine Science and Engineering Guangdong Laboratory (Zhuhai), China.

*Correspondence to*: Xi Liang (liangx@nmefc.cn)

**Abstract.** An operational synoptic scale sea ice forecasting system for the Southern Ocean, namely Southern Ocean Ice Prediction System (SOIPS), has been developed to support ship navigation in the Antarctic sea ice zone. Practical application of the SOIPS forecasts had been implemented for the 38th Chinese National Antarctic Research Expedition for the first time. The SOIPS is configured on an Antarctic regional sea-ice–ocean–ice-shelf coupled model and an ensemble-based Localized Error Subspace Transform Kalman Filter data assimilation model. Daily near-real-time satellite sea ice concentration observations are assimilated into the SOIPS to update sea ice concentration and thickness in the 12 ensemble members of model state. By evaluating the SOIPS performance on forecasting sea ice metrics in a complete melt-freeze cycle from October 1, 2021 to September 30, 2022, this study shows that the SOIPS can provide reliable Antarctic sea ice forecasts. In comparison with non-assimilated OSISAF data, annual mean root mean square errors of the sea ice concentration forecasts at lead time of up to 168-hour are lower than 0.19, and the integrated ice-edge errors of sea ice forecasts in most freezing months at lead times of 24-hour and 72-hour maintain around $0.5 \times 10^6$ km$^2$ and below $1.0 \times 10^6$ km$^2$, respectively. With respect to the scarce ICESat-2 observations, the mean absolute errors of the sea ice thickness forecasts at lead time of 24-hour are lower than 0.3 m, which is in range of the ICESat-2 uncertainties. Specifically, the SOIPS has a capacity in forecasting sea ice drift, both in magnitude and direction. The derived sea ice convergence rate forecasts have a high potential in supporting ship navigation on local fine scale. The comparison among the persistence forecasts, the SOIPS forecasts with and without data assimilation further shows that both model physics and data assimilation scheme play important roles in reliable sea ice forecasts in the Southern Ocean.

## 1 Introduction

Surrounding the Antarctica, sea ice motion in the Southern Ocean is fast. This situation is partly caused by the natural feature of Antarctic sea ice that the majority of the ice is thin first-year ice. Wind force leads to faster ice speed if ice thickness is thinner. Moreover, the severe Antarctic environmental conditions, such as frequent westerly cyclones, complicated surface ocean circulation system, drastic nighttime katabatic winds off the ice-shelf and coast, also promote the rapid ice motion. Beyond the Antarctic Peninsula, the topographic shape of high-latitude Southern Ocean without a land barrier in the zonal

direction provides an advantage for rapid sea ice movement (Worby et al., 1998; Heil and Allison, 1999; Turner et al., 2002; Wang et al., 2014; Womack et al., 2022). Energetic sea ice in the Southern Ocean has become one of the major challenges to safe maritime navigation due to the lack of timely and accurate sea ice forecasting information (Wagner et al., 2020), e.g. during austral summer of 2013/2014 both the Russian icebreaker *MV Akademik Shokalskiy* and the Chinese icebreaker *MV Xue Long* were trapped in the Adélie Depression region by quickly convergent sea ice under the influence of several cyclones (Witze, 2014; Turney, 2014; Zhai et al., 2015). Earlier in November 2007, a cruise ship *MS Explorer* sunk between the South Shetlands and Graham Land in the Bransfield Strait, after striking an iceberg near the South Shetland Islands, an area which is usually stormy but was calm at the time. Hence, reliable synoptic Antarctic sea ice forecasts are of great important to the Antarctic maritime commercial and scientific activities in the coming decades, when the human activities in the Southern Ocean are expected to be prosperous.

However, partly owing to the relative small amount of customers who needs Antarctic sea ice information, few attempts have been made by international weather forecasting centers to construct operational synoptic scale sea ice forecasting system for the Southern Ocean, in comparison to the multiple kinds of Arctic sea ice forecasting systems. The Canadian Meteorological Center (CMC) operates the Global Sea Ice Ocean Forecast System (GIOPS; Smith et al., 2016) which is built on the Nucleus for European Modelling of the Ocean (NEMO) version 3.1 and the Los Alamos National Laboratory Community Ice CodE (CICE) version 4.0, and the system is driven by atmospheric forcing from the Global Deterministic Prediction System. Since 2011, the GIOPS provides 10-day forecasts of global ocean and sea ice including the Southern Ocean at a resolution of 0.25°. The United Kingdom Met Office (UKMO) operates the Forecast Ocean Assimilation Model (FOAM; Blockley et al., 2014) which is also based on the NEMO and the CICE. Driven by atmospheric variables at ocean surface from the Met Office Unified Model (UM) global Numerical Weather Prediction (NWP) system, the FOAM produces 7-day forecasts of global ocean tracers, ocean currents and polar sea ice with a horizontal resolution of 0.25°. Under the framework of Copernicus Marine Environment Monitoring Service (CMEMS), the Mercator Ocean International (MOI) has developed a global ocean real-time monitoring and 1/12° high-resolution forecasting system (GLO-HR; Lellouche et al., 2018) based on the NEMO and the Louvain-la-Neuve Sea Ice Model version 2 (LIM2), and the atmospheric forcing is taken from the Integrated Forecast System (IFS). The GLO-HR delivers 10-day forecasts for global ocean and polar sea ice on a daily basis. The US Navy's Global Ocean Forecast System version 3.1 (GOFS 3.1) is based on the HYbrid Coordinate Ocean Model (HYCOM) and the CICE, and provides a global sea ice prediction capability including both the Arctic and the Antarctic (Posey et al., 2015). SEAS5, the fifth generation seasonal forecast system of the European Centre for Medium-Range Weather Forecasts (ECMWF), which is constituted by the NEMO ocean model, LIM2 sea ice model and IFS atmospheric model, has a horizontal resolution of 0.25° for global ocean and sea ice and provides 10-day forecasts of Antarctic sea ice cover and snow depth (Johnson et al., 2019). Nevertheless, all the above-mentioned operational forecasting systems are built on global coupled models, and their focus is not purely on Antarctic sea ice forecasts. Although resolution of global models is constantly becoming finer, regional ice–ocean coupled models at a similar resolution, but with lower

computational cost still offer some advantages when appropriate initial and boundary conditions are adopted (Mu et al., 2019; Liang et al., 2020; Ren et al., 2021).

Data assimilation is an essential way to reduce short-term forecast uncertainties by providing an optimal estimated initial state, which have been long-engaged in geophysical or biogeochemical applications (Verdy and Mazloff, 2017). Various data assimilation algorithms have been widely used to assimilate multi-source observations into the sea ice forecasting and

analysis systems (Lindsay and Zhang, 2006; Massonnet et al., 2013; Luo et al., 2021). Both the GIOPS and GLO-HR use System Assimilation Mercator version 2 (SAM2) as their ocean assimilation systems, which was developed from the Singular Evolutive Extended Kalman (SEEK) algorithm (Tranchant et al., 2006). The FOAM and SEAS5 adopt 3D-Var data assimilation systems for use with the NEMO, namely NEMOVAR (Mignac et al., 2022; Mogensen et al., 2009, 2012). The GOFS 3.1 employs the Navy Coupled Ocean Data Assimilation (NCODA) system based on 3D-Var method (Cummings and

Smedstad, 2014). The Southern Ocean State Estimate (Mazloff et al., 2010) constrains model state using in situ and satellite measurements through 4D-Var data assimilation. These systems mainly assimilate near-real-time satellite observations of sea ice concentration, sea level anomaly, sea surface temperature together with in situ observations of ocean temperature and salinity profiles. Previous studies have shown that the Ensemble Kaman Filter (EnKF) algorithm using dynamic background error covariance is suitable for multi-variable data assimilation in polar regions because it does not need to develop complex

adjoint models and is computationally efficient, thus it has been widely used in Arctic sea ice forecasts (Sakov et al., 2012; Yang et al., 2014, 2015, 2016; Mu et al., 2018; Liang et al., 2019).

In order to address the pressing need for sea ice forecasts in the Southern Ocean, especially in support of Chinese National Antarctic Research Expedition (CHINARE), the motivation of this work is to describe a newly developed regional synoptic scale forecasting system for Antarctic sea ice, i.e., Southern Ocean Ice Prediction System version 1.0 (SOIPS V1.0), which is

based on an sea-ice–ocean–ice-shelf coupled model and an EnKF data assimilation algorithm. The SOIPS operationally ran since January 1, 2021, and provided sea ice forecasts for the 38th CHINARE-Antarctic during austral summer of 2021/2022. Here, by evaluating sea ice forecasts in a complete melt-freeze cycle between October 1, 2021 and September 30, 2022, we show in this study that this new system has a capacity of providing precise forecasts for Antarctic sea ice evolution at synoptic timescale, especially the forecast accuracy of sea ice drift is substantially guaranteed. The paper is organized as

follows. In section 2, the system configuration, data assimilation strategy, and design of comparison experiments are described in detail. Antarctic sea ice forecasts, including sea ice concentration, sea ice edge, sea ice thickness, sea ice drift and sea ice convergence rate, are evaluated in section 3. Conclusions and discussions are made in section 4.

## 2 System Description

### 2.1 Model Configuration

The regional sea-ice–ocean–ice-shelf coupled model of SOIPS is configured on the Massachusetts Institute of Technology general circulation model (MITgcm; Marshall et al., 1997; Losch et al., 2010). The ocean model uses curvilinear coordinates with the open boundaries far north away from the domain of the Antarctic Circumpolar Current (ACC) and any likely northern extent of the sea ice. There are 496×496 grid points in horizontal with an average resolution of ~ 18 km (Figure 1). Vertically, it is composed of 50 unevenly spaced layers with intervals from 10 m near the surface to 450 m at the bottom.

The ocean model utilizing the finite-volume incompressible Navier-Stokes equations adopts the bulk formula for heat and momentum calculations at surface (Large and Pond, 1981, 1982) and the K-profile parameterization for vertical mixing in the ocean interior (KPP; Large et al., 1994). The viscous-plastic rheology (Hibler, 1979; Zhang and Hibler, 1997) and the zero-layer ice/snow thermodynamics (Semtner, 1976) are used in the sea ice model, which shares the same horizontal mesh with the ocean model. The ice-shelf, serving as a static surface boundary condition, exerts dynamic and thermodynamic

influences on the underlying ocean and thus affects ocean circulation and sea ice (Losch, 2008). Dynamically, the ice shelf draft on the top of the water column has a similar role as the surface orography. Underneath an ice shelf, the pressure at the top of the water column is the sum of the atmospheric pressure and the weight of the ice shelf column. Thermodynamically, the freezing and melting at the basal surface of the ice shelf can induce effective heat flux and virtual salt flux at the ice–ocean interface, with an additional tendency term of temperature and salinity to the ocean at the depth of the ice-shelf

draft. An oceanic boundary layer underneath the ice-shelf/ocean interface is formed following three physical constraints: the interface must be at the freezing point and both heat and salt must be conserved at the interface (Holland and Jenkins, 1999). Neither specific landfast ice parameterization, iceberg parameterization, nor tide forcing has been involved in the SOIPS. Time step of the coupled model is 1200 seconds.

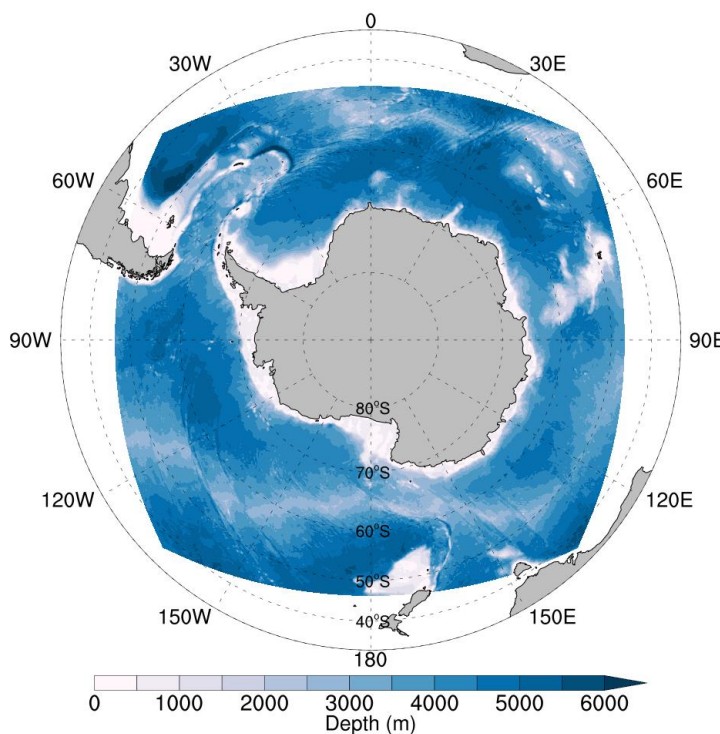

**Figure 1: The domain of the Southern Ocean Ice Prediction System (SOIPS). The contours show the bathymetry in meters.**

The initial fields of ocean temperature and salinity are derived from the World Ocean Atlas 2009 (WOA09; Locarnini et al., 2010; Antonov et al., 2010). The initial fields of sea ice concentration and thickness are obtained from observations of the Advanced Microwave Scanning Radiometer for the Earth Observing System (AMSR-E; Toudal Pedersen et al., 2017) and
the Ice, Cloud, and land Elevation Satellite (ICESat; Kurtz and Markus, 2012), respectively. The ice-shelf draft is obtained from a consistent data set of Antarctic ice sheet topography, cavity geometry, and global bathymetry (Timmermann et al. 2010). Climatological monthly mean oceanic boundary conditions are provided by the Estimating the Circulation and Climate of the Ocean phase II (ECCO2; Menemenlis et al., 2008), including ocean potential temperature, salinity, and velocity.

In our previous work, a model free run from 1979 to 2020 without data assimilation has been successfully conducted, which was forced by atmospheric variables at ocean surface derived from the Japanese 55-year Reanalysis (JRA55; Kobayashi et al., 2015; Harada et al., 2016) including 2-m air temperature and humidity, 10-m wind velocity components, downward shortwave and longwave radiation at the sea surface, and total precipitation. Validation of the model free run results including the simulated sea ice extent, sea ice concentration, sea ice thickness and net eastward oceanic volume transport
across the Drake Passage has demonstrated that this regional sea-ice–ocean–ice-shelf coupled model is capable in capturing the main features of Antarctic sea ice and ocean (Zhao et al., 2023).

## 2.2 Data Assimilation Scheme

The data assimilation algorithm used in SOIPS is the ensemble-based Localized Error Subspace Transform Kalman Filter (LESTKF, Nerger et al., 2012), which is packaged in the Parallel Data Assimilation Framework (PDAF; Nerger and Hiller, 2013). LESTKF is a localized variant of the Error Subspace Transform Kalman Filter (ESTKF), in which the dynamic background error covariance is applied. An optimal localization scheme that allows for adaptive localization radius based on observation number is achieved by setting the effective local observation dimension equal to the ensemble size (Kirchgessner et al., 2014). Weights of observations within the optimal localization radius are calculated based on a 5th-order polynomial function according to the distance between observation location and analysis grid point (Hunt et al., 2007; Gaspari and Cohn, 1999). Studies have indicated that LESTKF is suitable for high-dimensional models with small scale local features and large number of observations (Vetra-Carvalho et al., 2018). Considering the balance between computational efficiency and forecasting skills, 12 ensemble members are selected for the SOIPS ensemble forecasts.

SOIPS started running operationally on January 1, 2021. The initial ensemble of SOIPS was generated by disturbing the latest state of the model free run including sea ice concentration and thickness. The 12 perturbations which were used to initialize the SOIPS ensemble were created by applying an order-2 sampling scheme (Pham, 2001) to the leading 11 EOF modes of the daily model state evolution in the historical model free run between January 1, 2019 and December 31, 2020. During each assimilation step, near-real-time 6.25 km-resolution sea ice concentration data retrieved from the Advanced Microwave Scanning Radiometer 2 (AMSR2) brightness temperature data, were assimilated into the SOIPS and used to update sea ice concentration and thickness in the 12 ensemble initial fields on a daily basis. Since the uncertainties of the AMSR2 observations are not the same for different sea ice concentration ranges (Spreen et al., 2008), for simplicity a uniform value of 0.15 was assigned as the representative observation error. Specifically, a post-assimilation procedure was carried out that the modeled sea surface salinity is adjusted according to the formula described by Liang et al. (2019) to match with the change of sea ice thickness. Atmospheric forcing for operational forecasts were taken from the National Centers for Environmental Prediction (NCEP) Global Forecast System (GFS) 168-hour atmospheric forecasts. During each forecasting step, the 12 ensemble members after assimilating observed sea ice concentration were separately integrated for 168 hours, to create 12 members of 7-day forecasts and their ensemble mean was saved. The ensemble fields of the 24 h forecasts were also recorded as initial fields for the operational forecasts on the following day (Figure 2). In this study, we perform three experiments utilizing the SOIPS on a daily basis to disentangle the impact of data assimilation from that of model physics on the sea ice forecasts. The forecast experiment with data assimilation, denoted by DA_Forecast, assimilates the AMSR2 sea ice concentration data and is driven by the GFS data for 168 hours. The forecast experiment without data assimilation, denoted by NoDA_Forecast, is driven by the GFS data for 168 hours without any data assimilation. The persistence forecast experiment, denoted by PE_Forecast, uses the daily initial condition of the DA_Forecast run as forecasts of the following 168 hours. The operational SOIPS actually uses the setting of the DA_Forecast run, thus sea ice forecasts of

the DA_Forecast run are derived from the operational record of the SOIPS. The NoDA_Forecast and PE_Forecast runs have been conducted for comparison from October 1, 2021 and September 30, 2022.

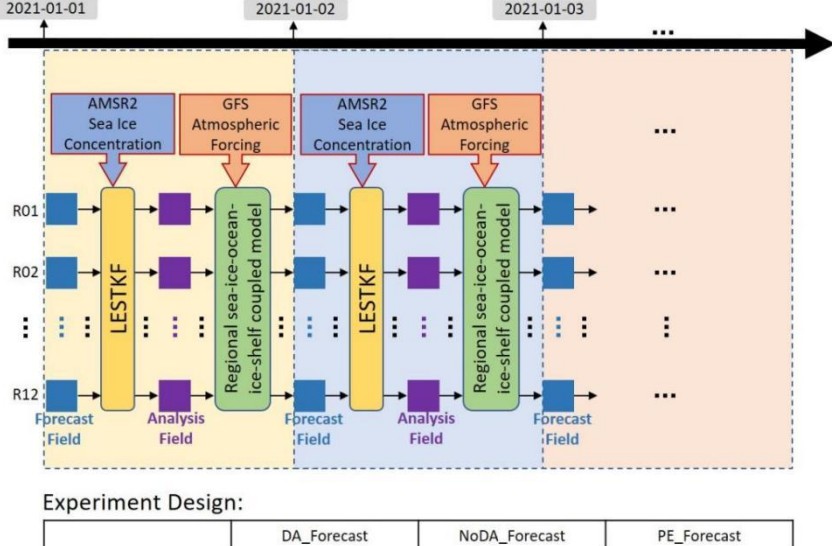

Figure 2: Schematic diagram of the SOIPS and the experiment design. The blue and purple squares denote the 12 ensemble members of model state pre- and post- data assimilation step. The yellow block denotes the data assimilation model utilizing the ensemble-based LESTKF. The green block denotes the Antarctic regional sea-ice–ocean–ice-shelf coupled model. The blue block with thick arrow denotes the near-real-time AMSR2 sea ice concentration observation. The orange block with thick arrow denotes the operational GFS atmospheric forcing.

## 3 Evaluation of Sea Ice Forecasts

SOIPS had provided forecasts of sea ice concentration, sea ice thickness, sea ice drift, sea ice convergence rate for the 38th CHINARE-Antarctic during the austral summer of 2021/2022. Here, we evaluate the three experiments during a complete melt-freeze cycle from October 1, 2021 to September 30, 2022. Additionally in the supplementary material, we also evaluate the operational records of the SOIPS until September 2023 to show that the SOIPS successfully predicted the historical Antarctic sea ice extent minima in 2023, and compare the SOIPS forecasts to the physical analysis field of the Antarctic Ocean produced by the MOI.

## 3.1 Sea Ice Concentration

The sea ice concentration product of the EUMETSAT Ocean and Sea Ice Satellite Application Facility (OSISAF), delivered daily at 10 km resolution in a polar stereographic projection, is used as an independent observation to evaluate the sea ice concentration forecasts. This product is computed from atmosphere-corrected brightness temperatures of the Special Sensor Microwave Imager/Sounder (SSMIS), using a combination of state-of-the-art algorithms which is different from the ARTIST Sea Ice (ASI; Spreen et al., 2008) algorithm used for AMSR2 sea ice concentration.

We calculate root mean square errors (RMSEs) between the SOIPS forecasts at different lead times and the OSISAF sea ice concentration observations, to evaluate the performance of SOIPS on sea ice concentration forecasts (Figure 3). As the spatial resolution of SOIPS is coarser than that of the OSISAF data, we interpolate the OSISAF data onto the model grid of SOIPS. Basically the RMSEs of the DA_Forecast run at each lead time gradually increase during October–March (hereafter the latter month in such expressions that the latter month is earlier than the former month denotes the month of the next year)

followed by a decrease starting from April. The RMSEs of the DA_Forecast run are generally lower than 0.15 during June–September while close to 0.2 during January–February. The RMSEs of the DA_Forecast run has two peaks, one in December and the other in April. The maximum RMSE of the DA_Forecast run in April is lower than 0.33. Comparison of the DA_Forecast run at different lead times shows that the RMSEs increase generally along with the prolong of forecast lead time. Statistical analysis reveals that annual mean RMSEs of the DA_Forecast run at lead times of 24-hour, 72-hour, 120-

hour and 168-hour are 0.15, 0.16, 0.17 and 0.19, respectively. Comparison of the three experiments shows that the DA_Forecast run performs best and the NoDA_Forecast run performs worst in most time except during late March–early June. Since the PE_Forecast run includes observed sea ice concentration information, the PE_Forecast run generally performs better than the NoDA_Forecast run and worse than the DA_Forecast run. During late March–early June, the PE_Forecast run performs worse than the other two runs at lead time of 168-hour, suggesting that sea ice changes rapidly in

response to the oceanic and atmospheric forcing during this onset–to–fast freezing period. We also assess the difference between the assimilated AMSR2 and the OSISAF sea ice concentration data. Due to different remote sensors and retrieval algorithms, there are significant systematic deviations between the OSISAF and AMSR2 products. The RMSEs of these two products increase in the melting season reaching a maximum value of 0.24 in February, thereafter the RMSEs decrease rapidly in April maintaining below 0.15 in the rest of the freezing season. The systematic bias between the assimilated data

and the validation data partly explains the sea ice concentration forecasting errors.

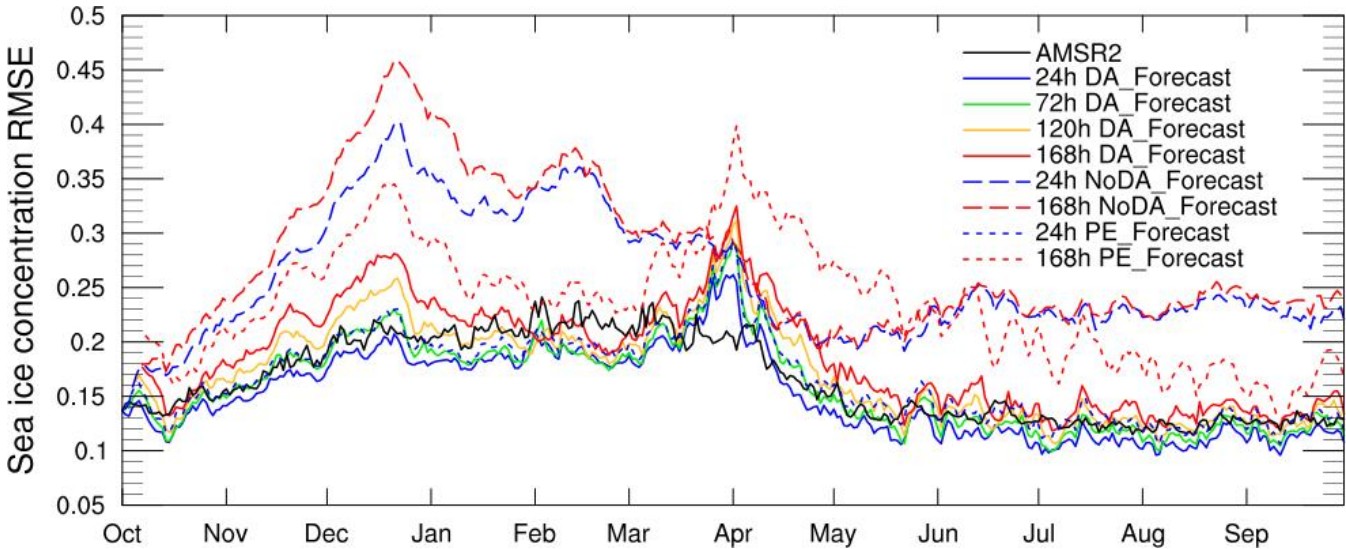

**Figure 3: Time series of the RMSEs of the assimilated AMSR2 data and sea ice concentration forecasts at different lead times with respect to the OSISAF data. The blue, green, yellow, red, and black solid lines denote the sea ice concentration forecasts of the DA_Forecast run at lead times of 24-hour, 72-hour, 120-hour, 168-hour, and the AMSR2 data, respectively. The blue and red long-dashed lines denote the sea ice concentration forecasts of the NoDA_Forecast run at lead times of 24-hour and 168-hour, respectively. The blue and red short-dashed lines denote the sea ice concentration forecasts of the PE_Forecast run at lead times of 24-hour and 168-hour, respectively.**

We further analyze spatial distributions of sea ice concentration forecasting errors by evaluating monthly mean fields of the DA_Forecast run at lead time of 24-hour (Figure 4). During October–November, relative large RMSEs of sea ice concentration forecasts are mainly located in the north marginal ice zone surrounding the Antarctica, where the sea ice, normally with relative low concentration and thickness, moves actively in response to external forces. In December, the RMSEs of sea ice concentration forecasts in the marginal ice zone greatly shrink except that in the Southern Atlantic Ocean sector between 30°W and 30°E. During January–February, the sea ice concentration forecasting errors are small in entire ice zone except in some nearshore areas of the eastern Antarctic. The sea ice concentration forecasting errors start to increase in the Ross–Amundsen Seas along with the northward expansion of sea ice zone during March–April. In the following freezing months, relative large RMSEs of sea ice concentration forecasts reemerge in the north marginal ice zone but their amplitudes are lower than those in the previous October–November. The monthly patterns of the RMSEs between the AMSR2 and OSISAF data (Figure S1 in the supplementary material) resemble and set the base for those between the DA_Forecast run and OSISAF data.

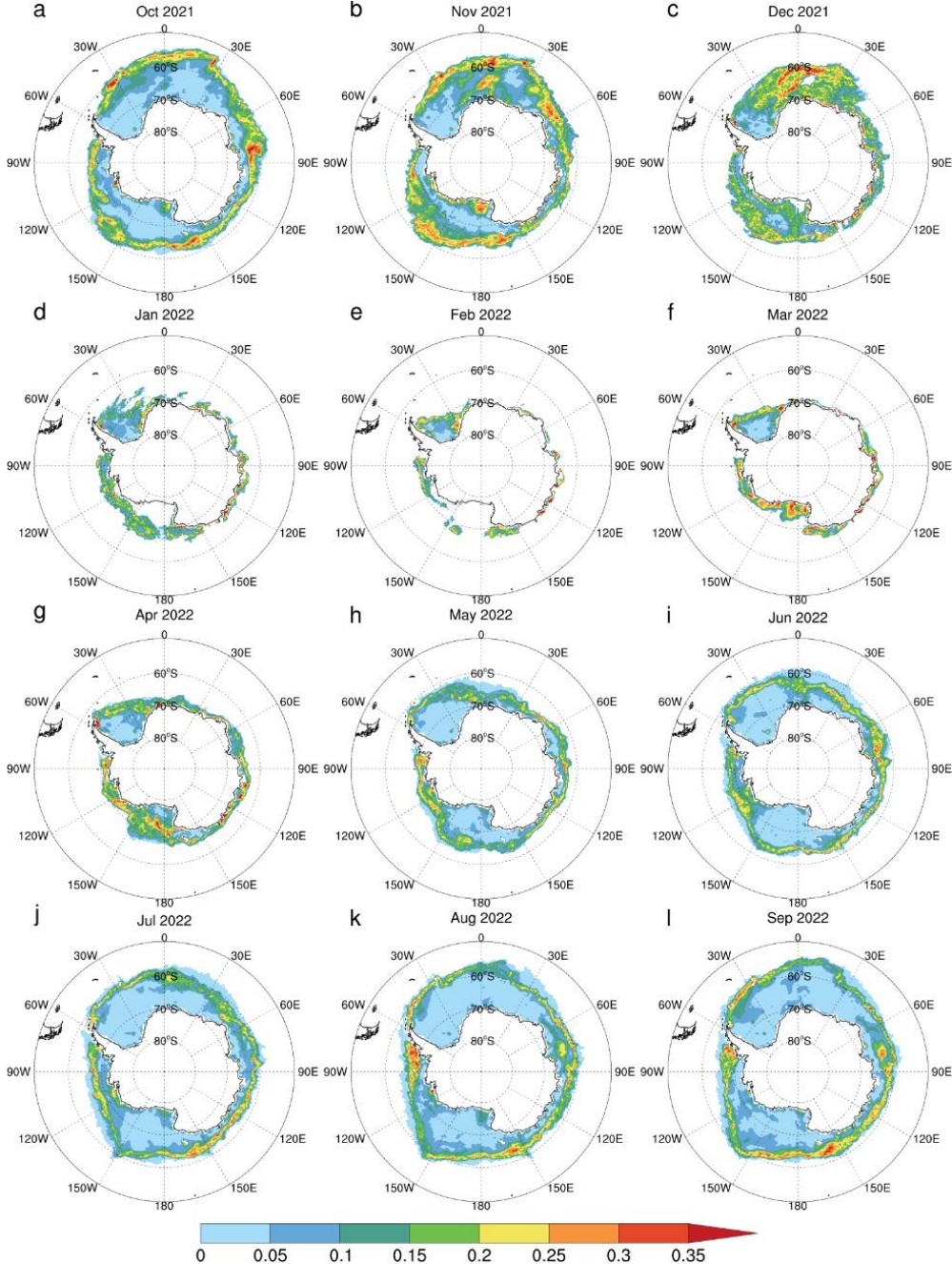

**Figure 4: Monthly patterns of the sea ice concentration RMSEs of the DA_Forecast run at lead time of 24-hour with respect to the OSISAF data. (a)–(l) denote October 2021–September 2022.**

**3.2 Sea Ice Edge**

Instead of evaluating just a sea ice extent number, Goessling et al. (2016) has introduced a more useful verification metric, i.e. Integrated Ice-Edge Error (IIEE), which is the sum of all areas where the local sea ice extent is overestimated or underestimated. Here sea ice edge is defined as the locations where sea ice concentration is 15%. Firstly, we evaluate the derived sea ice edges from the assimilated AMSR2 and the OSISAF data (Figure 5). The IIEEs are larger than $0.5 \times 10^6$ km$^2$ during October–early January, and smaller than $0.5 \times 10^6$ km$^2$ in other months. The maximum IIEE occurs in December with

a value of $1.45 \times 10^6$ km$^2$. The sea ice edge biases between the assimilated AMSR2 and the OSISAF data contribute to the first peak in the sea ice concentration RMSEs of the DA_Forecast run in December as shown in Figure 3.

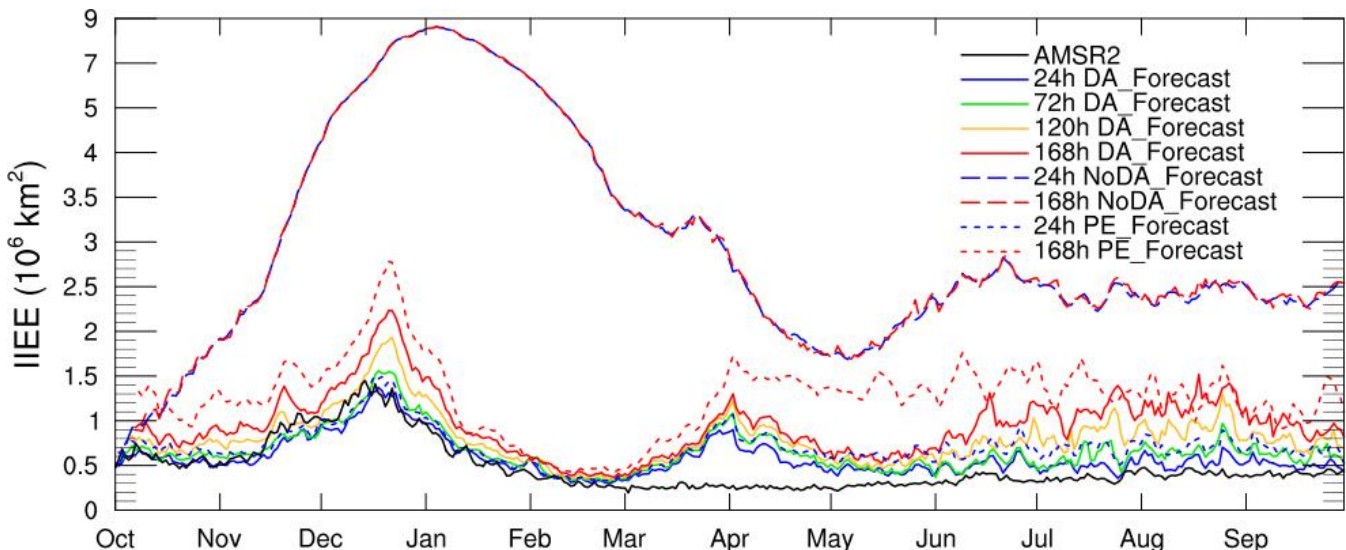

**Figure 5: Time series of the IIEEs of the assimilated AMSR2 data and the forecasts at different lead times with respect to the OSISAF data. The blue, green, yellow, red, and black solid lines denote the IIEEs of the DA_Forecast run at lead times of 24-hour,**

**72-hour, 120-hour, 168-hour, and the AMSR2 data, respectively. The blue and red long-dashed lines denote the IIEEs of the NoDA_Forecast run at lead times of 24-hour and 168-hour, respectively. The blue and red short-dashed lines denote the IIEEs of the PE_Forecast run at lead times of 24-hour and 168-hour, respectively.**

The evolutions of the IIEEs of the DA_Forecast run at different lead times have similar shapes to that of the assimilated

AMSR2 data. In December the maximum IIEEs of the DA_Forecast run at different lead times range from $1.35 \times 10^6$ km$^2$ to $2.25 \times 10^6$ km$^2$. In the early freezing season, large IIEEs of the DA_Forecast run reemerge at the end of March, corresponding to the second peak in the sea ice concentration RMSEs of the DA_Forecast run (Figure 3). The large IIEEs of the DA_Forecast run in late March and early April can not be attributed to the sea ice edge biases between the assimilated AMSR2 and the OSISAF data, but rather the model ability in accurately simulating the expansion of sea ice cover in the

early freezing season. During June–September, the IIEEs of the DA_Forecast run at lead time of 24-hour maintain around

$0.5 \times 10^6 \, \text{km}^2$, and those of 72-hour are below $1 \times 10^6 \, \text{km}^2$. Comparison of the three experiments on sea ice edge forecasts shows that the DA_Forecast run performs best and the NoDA_Forecast run performs worst over the whole study period.

Spatially at a first glance, the sea ice edge forecasts of the DA_Forecast run at lead time of 24-hour are generally coincident with those in the OSISAF data (Figure 6). The sea ice edge forecasting biases of the DA_Forecast run at lead time of 168-hour (Figure S2 in the supplementary material) grow obviously in November−December, March−April, and July−August. The areas with large sea ice edge biases are located in the southeastern Atlantic Ocean sector, southwestern Indian Ocean sector, and southwestern Pacific Ocean sector. It is noteworthy that besides the contributor to the IIEEs from the north marginal ice zone, significant contributor to the IIEEs is from the nearshore areas surrounding the Antarctica in all months. By carefully checking the coastlines or ice-shelf fronts of the Antarctica in the model domain and in the OSISAF data, we realize that part of mismatch of sea ice edges in the nearshore areas is mendacious originating from the divergence of coastlines or ice-shelf fronts in the model domain and in the OSISAF data. The lack of specific landfast ice parameterization may lead to unrealistic landfast ice zones around the Antarctica, which possibly also contribute to the mismatch of sea ice edges. The real IIEEs between sea ice forecasts and the OSISAF data should be lower than those in Figure 5.

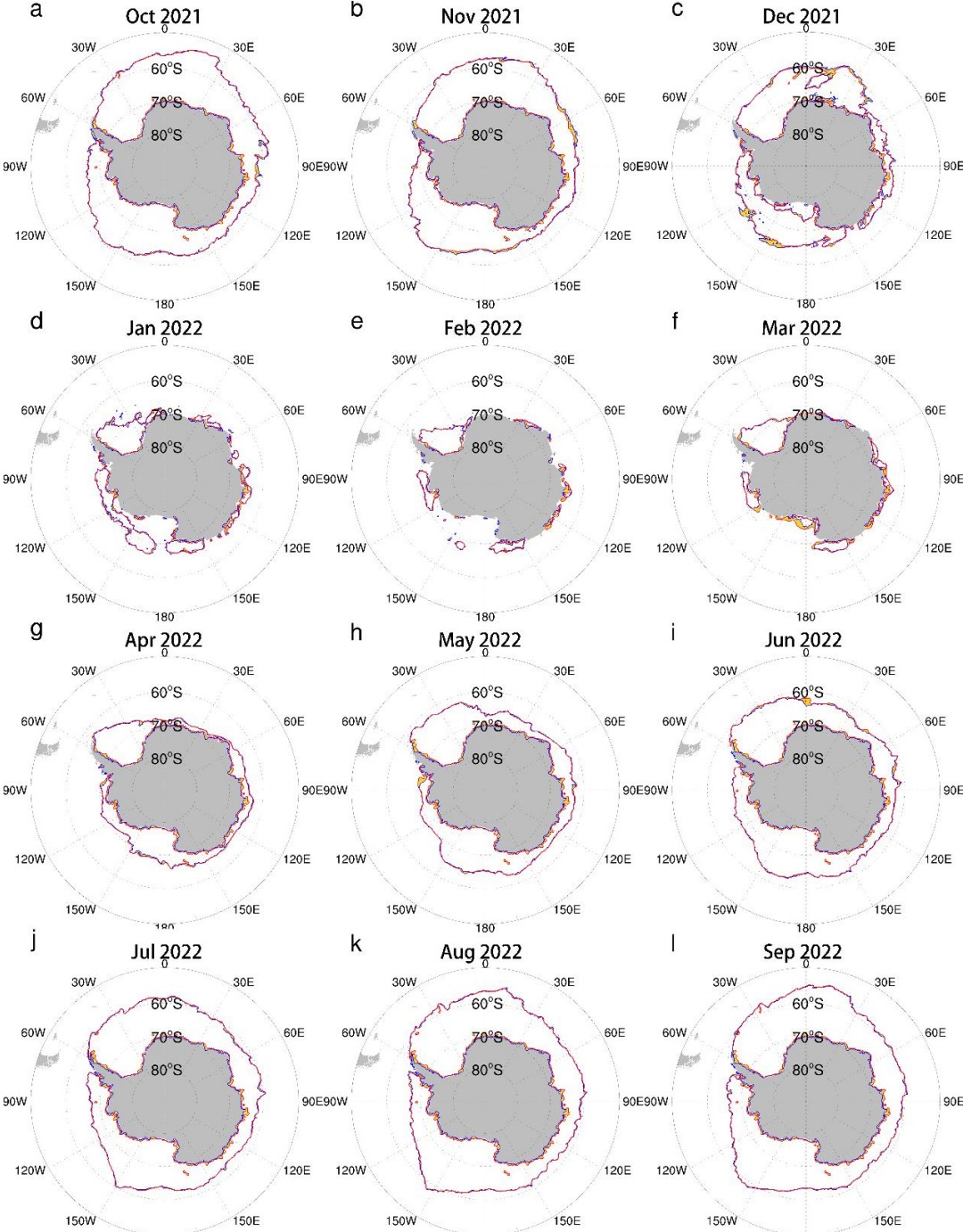

Figure 6: Monthly patterns of sea ice edge forecasts at lead time of 24-hour with respect to the OSISAF data. (a)-(l) denote October 2021–September 2022. The blue lines denote the DA_Forecast run. The red lines denote the OSISAF data. The gold contours denote the mismatch between these two data.

### 3.3 Sea Ice Thickness

At present, continuous observations of the Antarctic sea ice thickness over large area are still difficult to obtain. With the launch of Ice, Cloud and Land Elevation Satellite-2 (ICESat-2) on September 15, 2018, the Antarctic sea ice freeboard can be estimated from measurements of the Advanced Topographic Laser Altimeter System (ATLAS) instrument. By applying the improved One-Layer Method (OLMi, Xu et al., 2021) to the daily gridded sea ice freeboard estimate product ATLAS/ICESat-2 L3B, we obtain daily Antarctic sea ice thickness distribution at discrete locations from October 1, 2021 to
September 30, 2022.

We validate the daily evolution of the mean sea ice thickness forecasts at the discrete locations where observations on the corresponding date are available (Figure 7). The results show that sea ice thickness forecasts of the DA_Forecast run at lead time of 24-hour are consistent with the ICESat-2 observations basically, with an overestimation during October–November. In most time of the validation period, the mean absolute errors (MAEs) of the sea ice thickness forecasts are lower than 0.3
m, which is significantly smaller than the uncertainties of the ICESat-2 observations. Although without assimilation of sea ice thickness data, the DA_Forecast run also performs better than the NoDA_Forecast run on sea ice thickness forecasts. Sea ice thickness forecasts of the PE_Forecast run (not shown) are approximate equal to those of the DA_Forecast run at lead time of 24-hour since sea ice thickness changes a relatively small amount in one day.

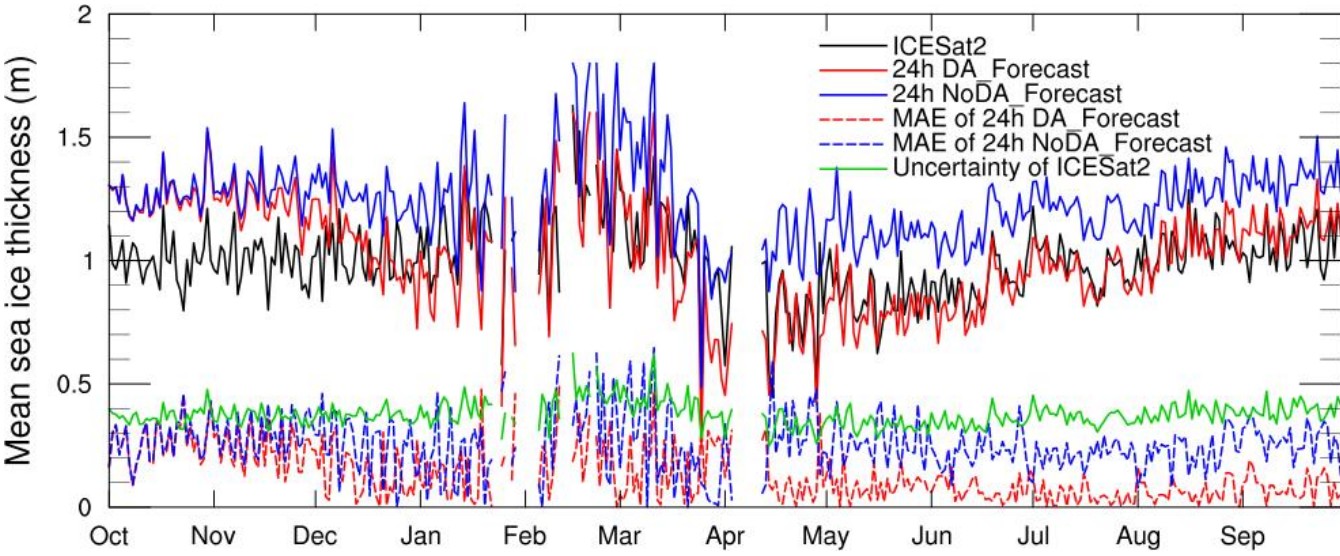

**Figure 7: Time series of the mean sea ice thickness and uncertainties of the ICESat-2 observations (black and green lines), the sea ice thickness forecasts at lead time of 24-hour in the DA_Forecast and NoDA_Forecast runs (red and blue solid lines), and the mean absolute errors between the forecasts and observations (red and blue dashed lines).**

We further evaluate spatial patterns of sea ice thickness forecasts of the DA_Forecast run at lead time of 24-hour (Figure 8)

after merging the daily sea ice thickness observations into seasonal mean fields. The sea ice thickness forecasts show a good agreement with the observations, featured by thick ice located in the Weddell Sea, the Amundsen Sea and the nearshore areas of the eastern Antarctic. During January–March, the DA_Forecast run overestimates ice thickness in the southern Weddell Sea while underestimates ice thickness in the eastern Amundsen Sea. In other seasons, the DA_Forecast run overestimates ice thickness in the western Ross Sea and the southern Weddell Sea, while underestimates ice thickness in the

Amundsen Sea and the nearshore areas of the eastern Antarctic. The forecasting errors in the southern Weddell Sea are in the range of the ICESat-2 uncertainties, but the forecasting errors in the western Ross Sea are out of the range of the ICESat-2 uncertainties. We suspect that the larger sea ice thickness biases in these areas are caused by the poor simulation of growth rate of sea ice thickness during the freezing seasons, partly originating from the biases in the simulated ocean temperature or air temperature in the GFS data. The biases of sea ice thickness forecasts of the DA_Forecast run at lead time of 168-hour

(Figure S3 in the supplementary material) do not change obviously in comparison with those of 24-hour. Admittedly, the above evaluation ignores the errors caused by the spatiotemporal discontinuity and the uncertainties of the ICESat-2 observations.

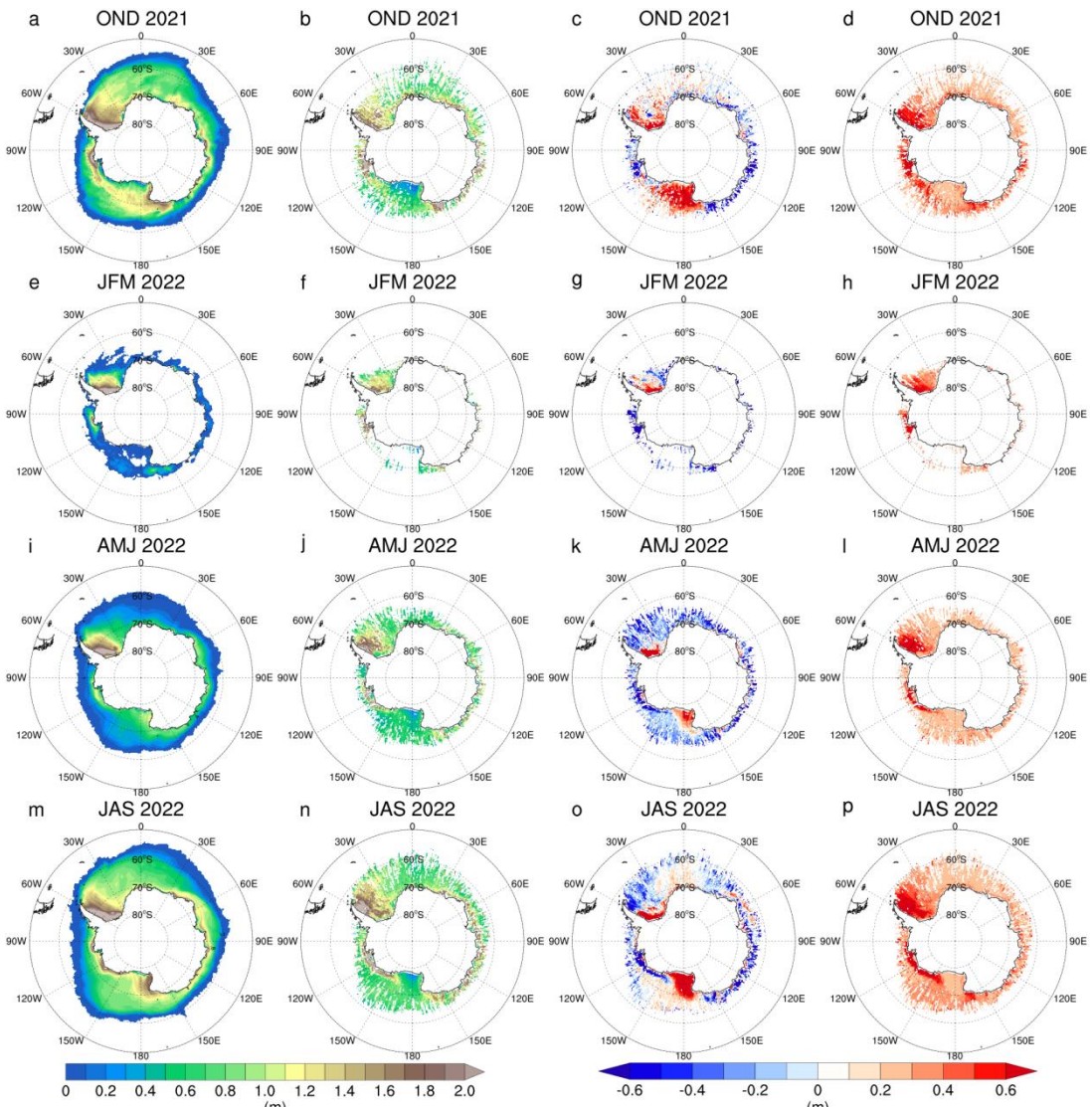

**Figure 8: Seasonal patterns of the Antarctic sea ice thickness. The columns from left to right denote the DA_Forecast run at lead time of 24-hour, the ICESat-2 observations, their deviations, and the uncertainties of the ICESat-2 observations, respectively. The panels from top to bottom denote October–December, January–March, April–June, and July–September, respectively.**

### 3.4 Sea Ice Drift

Polar pathfinder daily Antarctic sea ice motion product provided by the National Snow and Ice Data Center (NSIDC, Tschudi et al., 2019) is used to assess the Antarctic sea ice drift forecasts. This dataset is projected on the EASE grid with a spatial resolution of 25 km, including input data sources derived from the Advanced Very High Resolution Radiometer

(AVHRR), AMSR-E, Scanning Multichannel Microwave Radiometer (SMMR), Special Sensor Microwave/Imager (SSM/I), and SSMIS sensors, the International Arctic Buoy Programme (IABP) buoys, and the National Centers for Environmental Prediction/National Center for Atmospheric Research (NCEP/NCAR) Reanalysis.

To validate sea ice drift forecasts, we convert the NSIDC ice drift components $(u_o, v_o)$ on the EASE coordinates into the ice drift components $(u_m, v_m)$ on the model coordinates. Sea ice drift direction, expressed by the angle $\alpha$ with reference to location-dependent coordinate of $u_m$, is derived as the four quadrant arctangent of $(u_m, v_m)$. Note that $\alpha$ ranges between -180° and 180°. Sea ice drift direction bias is represented by the MAE of $\alpha$ between the modeled and observed sea ice drift. Sea ice drift magnitude is independent of selected coordinates.

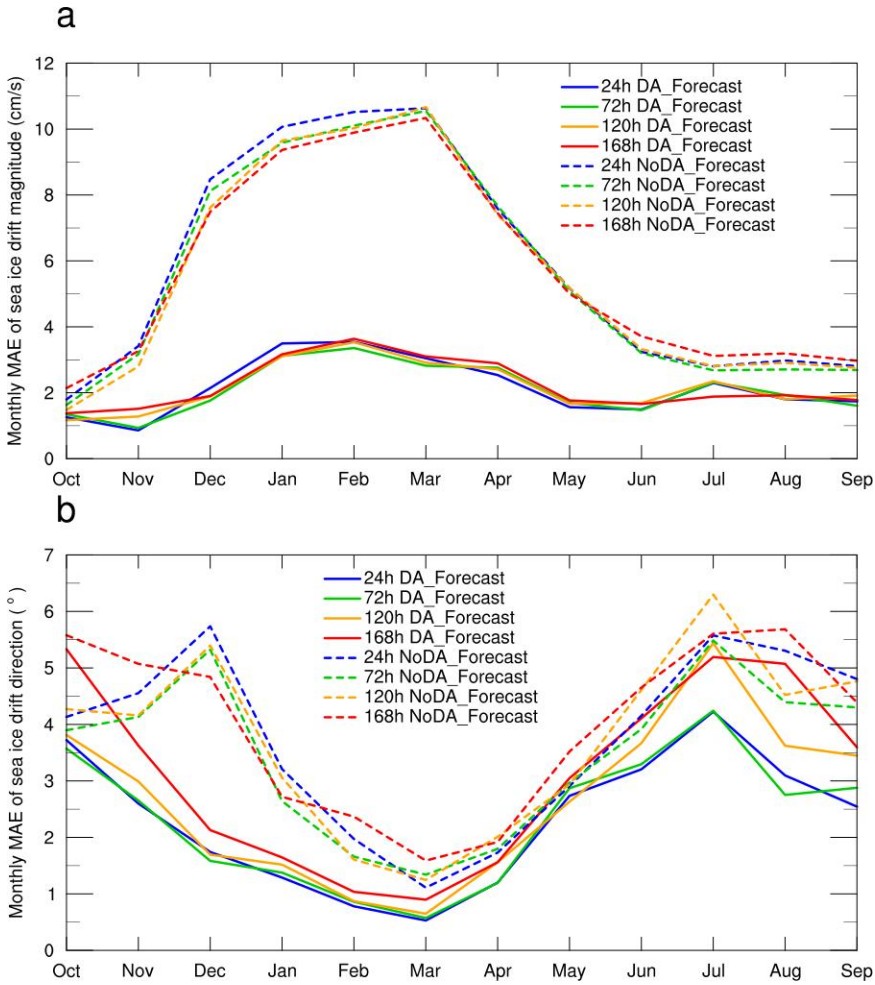


**Figure 9: Time series of the monthly-mean MAEs of (a) magnitude and (b) direction of the sea ice drift forecasts at different lead times with respect to the NSIDC data. The blue, green, yellow, and red lines denote the sea ice drift forecasts at lead times of 24-hour, 72-hour, 120-hour, and 168-hour, respectively. The solid and dashed lines denote the DA_Forecast and NoDA_Forecast runs, respectively.**

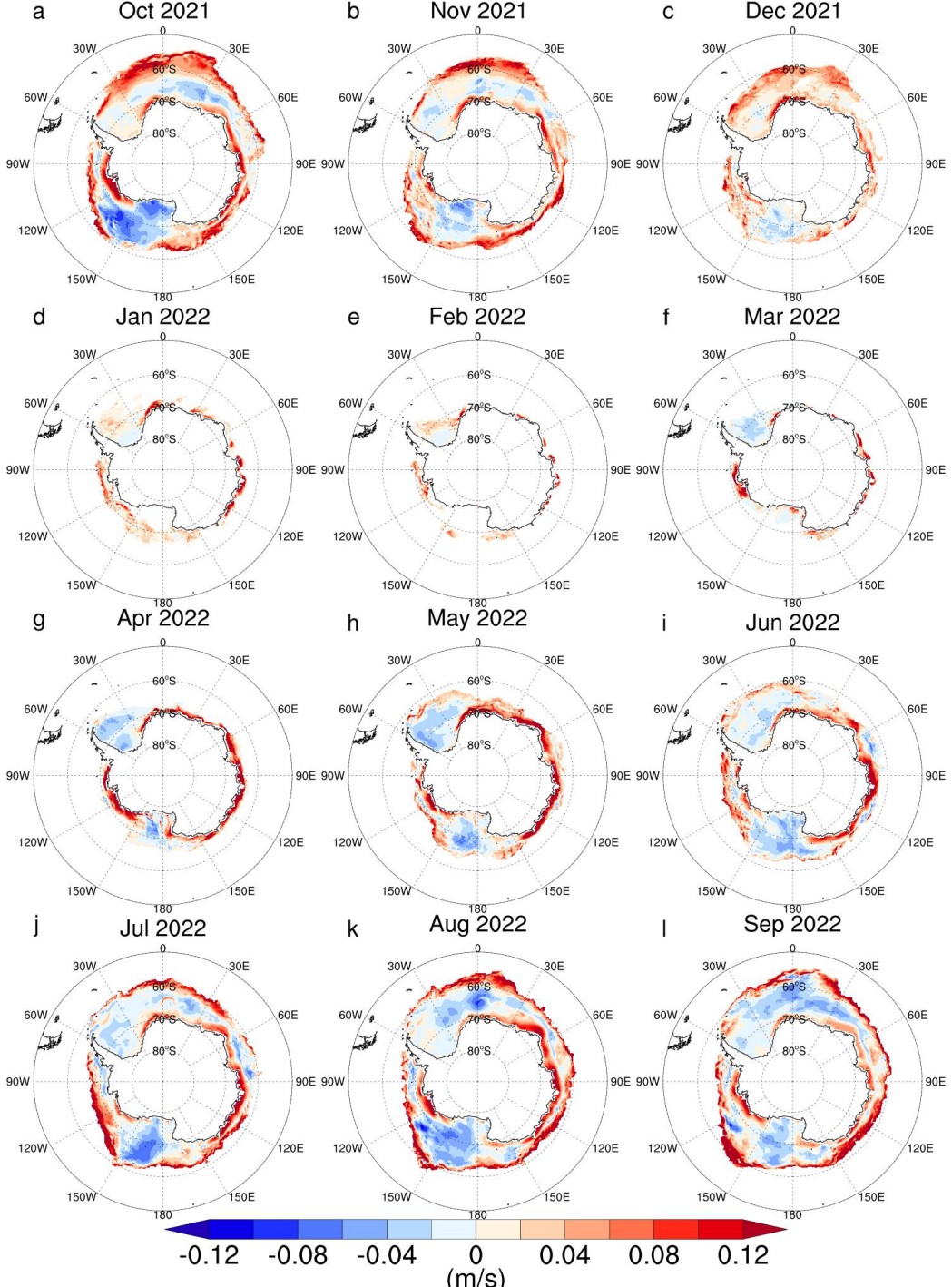


**Figure 10. Monthly patterns of the magnitude bias of sea ice drift between the DA_Forecast run at lead time of 24-hour and the NSIDC data. (a)-(l) denote October 2021–September 2022.**

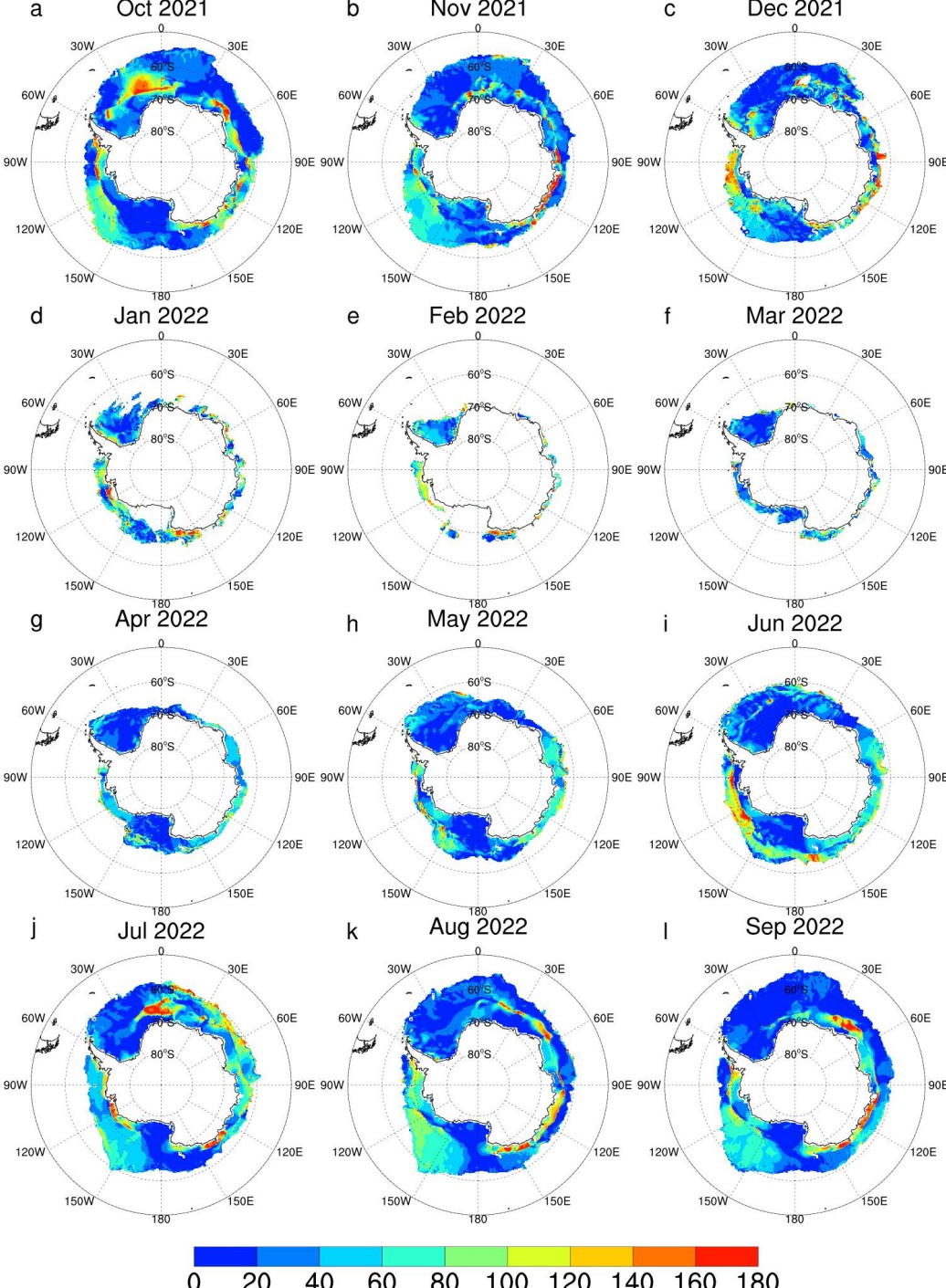

**Figure 11. Monthly patterns of the direction bias of sea ice drift between the DA_Forecast run at lead time of 24-hour and the**
**NSIDC data. (a)-(l) denote October 2021–September 2022.**

Validation results (Figure 9) show that the MAEs of sea ice drift magnitude between the DA_Forecast run and observations increase during November–February and decrease during March–May. In contrast, the MAEs of sea ice drift direction between the DA_Forecast run and observations decrease during October–February and increase during March–July.

Comparison between the DA_Forecast and NoDA_Forecast runs shows that the DA_Forecast run performs better than the NoDA_Forecast run, both on magnitude and direction of sea ice drift forecasts. The improvement of sea ice drift forecasts originates in principle from the enhancement of the SOIPS forecasts on sea ice concentration and thickness induced by data assimilation of the observed sea ice concentration, since sea ice drift is impacted by both sea ice concentration and thickness (Leppäranta, 2011).

As the forecast lead time increases, the MAEs of the sea ice drift magnitude do not exhibit significant amplification, but those of direction grow significantly at lead time of 168-hour in October–November and June–September. Statistical analysis (Table 1) shows that in the DA_Forecast run, the annual mean forecasting errors of sea ice drift magnitude at lead times of 24-hour, 72-hour, 120-hour, and 168-hour are 2.14 cm/s, 2.09 cm/s, 2.17 cm/s, and 2.22 cm/s, respectively. As a reference, the derived NSIDC sea ice drift magnitudes are 10.22 cm/s during October–December, 4.78 cm/s during January–March,

10.55 cm/s during April–June, and 13.26 cm/s during July–September. The annual mean forecasting errors of sea ice drift magnitude at lead time of 168-hour accounts for 23% of the observed magnitude. The annual mean forecasting errors of sea ice drift direction at lead times of 24-hour, 72-hour, 120-hour, and 168-hour are 2.13°, 2.08°, 2.42°, and 2.81°, respectively. This results suggest that the SOIPS has a reliable performance on forecasting sea ice drift direction, although with a systematic positive bias in the sea ice drift magnitude. A previous study conducted for the Arctic region has also found that

the numerical overestimation of sea ice drift speed is a common feature in the CMIP6 models (Wang et al., 2023).

Spatially, the DA_Forecast run at lead time of 24-hour produces larger sea ice drift magnitude in the north marginal sea ice zone and the coastal areas, while in between the DA_Forecast run produces smaller sea ice drift magnitude (Figure 10). During January–March, the Antarctic sea ice zone shrinks to its annual minima, sea ice drift magnitude bias appears to be relatively small compared to the other months. While in other months, large biases in sea ice drift direction forecasts also

occur in the densely packed sea ice zone, especially the Bellingshausen-Amundsen-Ross Seas and the southeastern Antarctic Ocean sector (Figure 11), thus the MAEs of sea ice drift direction forecasts are large.

| | | Forecast lead time | | | |
| --- | --- | --- | --- | --- | --- |
| | | 24h | 72h | 120h | 168h |
| MAEs of magnitude of sea ice drift (cm/s) | OND | 1.42 | 1.35 | 1.45 | 1.61 |
| | JFM | 3.36 | 3.09 | 3.17 | 3.29 |
| | AMJ | 1.86 | 1.97 | 2.02 | 2.10 |
| | JAS | 1.95 | 1.95 | 2.02 | 1.86 |
| | Average | 2.14 | 2.09 | 2.17 | 2.22 |
| MAEs of direction of sea ice drift (°) | OND | 2.59 | 2.40 | 2.57 | 3.38 |
| | JFM | 0.79 | 0.85 | 0.93 | 1.11 |
| | AMJ | 2.06 | 2.15 | 2.28 | 2.58 |
| | JAS | 3.07 | 2.91 | 3.89 | 4.16 |
| | Average | 2.13 | 2.08 | 2.42 | 2.81 |

**Table 1. The seasonal-mean MAEs of magnitude and direction of the DA_Forecast run at different lead times with respect to the NSIDC data.**

## 3.5 Sea Ice Convergence Rate

Sea ice convergence rate (SICR), defined as $SICR = -(\partial u_m / \partial x + \partial v_m / \partial y)$ (negative value represents sea ice dispersion, positive value represents sea ice accumulation), is a useful metric in guiding ship navigation in sea ice zone. The Antarctic China Zhongshan Station is located at (69°22'24.76"S, 76°22'14.28"E) in the Prydz Bay (Figure 12). In the southern Prydz Bay there is a large area of landfast ice. Drifting sea ice occupies the area north of the landfast ice zone. Under the forces of wind and tide, the drifting sea ice zone sometimes adhere to the landfast ice zone closely, sometime keep away from the landfast ice zone creating an open water band between them.

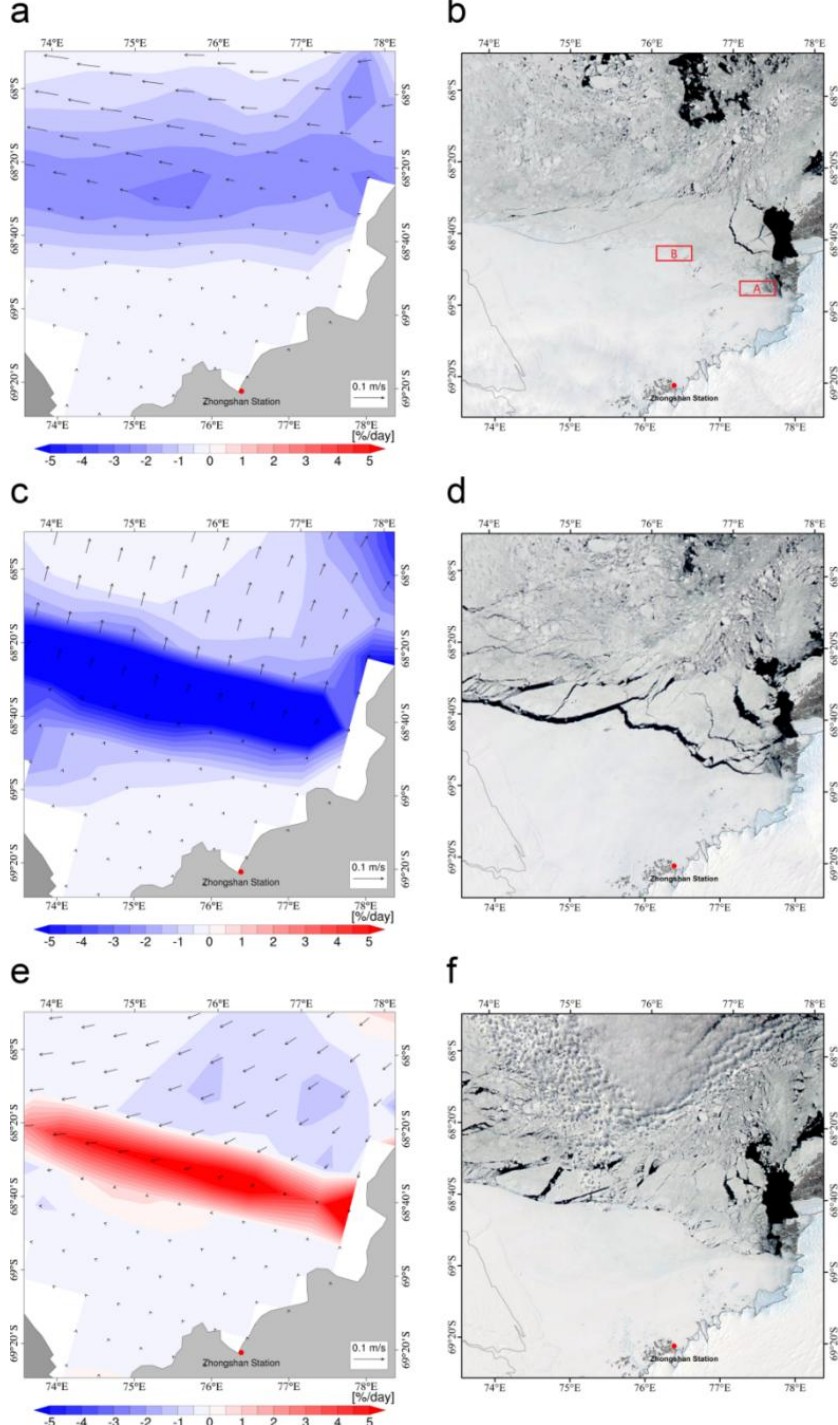

**Figure 12: Sea ice convergence rate in the DA_Forecast run (left column) and MODIS satellite images (right column). The top, middle, and bottom panels denote forecasts/observations on November 19, 20, and 21, 2021 respectively. The forecast was**

 **initialized on November 18, 2021. Black arrows in the left column denote sea ice drift vectors, while red and blue contours indicate that sea ice drift in the corresponding area tends to convergent and divergent, respectively. The red dot in each figure marks the Antarctic China Zhongshan Station. The two boxes in (b) denotes two areas where the icebreaker *MV Xue Long* has arrived in some years.**

The Chinese icebreaker *MV Xue Long* navigated to the Antarctic China Zhongshan Station to unload supplies almost every year in the past four decades. In some years, the icebreaker navigated southward to arrive the area of A through the relative loose drifting sea ice zone in the eastern Prydz Bay. However, owing to the indurative ice condition with many ice ridges and neaped icebergs in the landfast ice zone south of the area of A, the icebreaker had to navigate to the area of B and then turned southward heading to the Antarctic China Zhongshan Station. The landfast ice condition in the areas south of the area

of B is much friendly to the icebreaker. As a consequence, the timing of open water band between the drifting sea ice zone and the landfast ice zone plays a crucial role in the icebreaker navigation from A to B.

Here we show a typical situation of how the sea ice convergence rate benefits for the navigation from A to B. Forecasting initialized on November 18, 2021, the DA_forecast run at lead times of 24-hour, 48-hour, and 72-hour suggested a weak sea ice dispersion on November 19, 2021, a strong sea ice dispersion on November 20, 2021, and a strong sea ice accumulation

on November 21, 2021. The ice convergence rate forecasts indicated the open water band between the drifting sea ice zone and the landfast ice zone may occur on November 19, 2021, very likely to occur on November 20, 2021, and may disappear on November 21, 2021. The NASA MODIS images in the three days clearly validate the usability of the sea ice convergence rate forecasts during this opening-closing process of open water band. Further analysis shows that the forecasting skill of sea ice convergence originates largely from the precise atmosphere forcing rather than the effects of sea ice concentration data

assimilation (not shown).

## 4 Conclusion and Discussion

In this work we introduce an operational synoptic scale sea ice forecasting system for the Southern Ocean, i.e. Southern Ocean Ice Prediction System (SOIPS). The system is developed to meet the increasing demands for synoptic scale Antarctic sea ice forecasts at present and in the coming decade. The system is configured on an Antarctic regional sea-ice–ocean–ice-

shelf coupled model and an ensemble-based LESTKF data assimilation model, and driven by operational atmospheric forecasting variables at ocean surface from international weather forecasting products. Near-real-time satellite sea ice concentration observations are assimilated into the system on a daily basis to update sea ice concentration and thickness in the 12 ensemble members of model state. The SOIPS forecasts has been engaged in sea ice service for the 38th Chinese National Antarctic Research Expedition for the first time.

By evaluating sea ice forecasts in a complete melt-freeze cycle between October 1, 2021 and September 30, 2022, this study finds that the SOIPS has a reliable ability to forecast sea ice evolution on synoptic scale. With respect to the OSISAF data,

the sea ice concentration RMSEs of the SOIPS forecasts at lead time of up to 168-hour are generally lower than 0.15 during June–September while close to 0.2 during January–February, and the annual mean RMSEs are lower than 0.19. Relative large RMSEs are mainly located in the north marginal ice zone surrounding the Antarctica. The AMSR2 sea ice
concentration data are assimilated into the ensemble of model restart fields on a daily basis, and an analyzed (updated) ensemble of model restart fields combining the modeled and observational sea ice states are generated, which are further integrated for 168 hours driven by atmospheric forcing. The forecasts include not only the observational information, but also sea ice changes generated by model physics. This causes the smaller sea ice concentration RMSEs of the SOIPS forecasts in comparison with that of the AMSR2 data, especially at lead times of 24-hour and 72-hour in January–early
March and May–September. On the other side, large sea ice concentration RMSE appears in most areas of sea ice zone around the Antarctica in March–April, suggesting that the model has a relative low capacity in correctly simulating sea ice growth rate during this onset–to–fast freezing period. This probably originates from that the sea ice model in the SOIPS uses the zero-layer ice/snow thermodynamics, which is a simple one compared to sophisticated multi-layer ice/snow thermodynamics. Additionally as a reference, the sea ice concentration RMSE of the GOIPS forecasts at lead time of 168-
hour maintains below 0.35 in the year of 2011 with respect to the Interactive Multisensor Snow and Ice Mapping System ice extent product (Helfrich et al., 2007). With respect to the OSISAF data, the sea ice concentration RMSE of the SOIPS forecasts at lead time of 24-hour is larger than that of the MOI product. It should be mentioned that the MOI product has assimilated the OSISAF sea ice concentration data, which leads to a lower RMSE in comparison to the SOIPS forecasts (Figure S4 in the supplementary material).
The IIEEs of the SOIPS forecasts in most freezing months at lead times of 24-hour and 72-hour maintain around $0.5 \times 10^6$ km$^2$ and below $1.0 \times 10^6$ km$^2$, respectively. In comparison with July–December, the sea ice zone is smaller during January–June, so the IIEE grows moderately in response to prolonged forecast lead time. Moreover, the sea ice edge locates more north during July–December, and the marginal ice zone is more close to the ACC-impacting areas where active oceanic and atmospheric dynamical processes promote the amplification of the IIEE along with the prolong of forecast lead time. It
should be mentioned that mismatch of sea ice edges in some nearshore areas originates from the divergence of coastlines, ice-shelf fronts or unrealistic landfast ice zones in the model domain and the OSISAF data.

The sea ice thickness MAEs between the SOIPS forecasts at lead time of 24-hour and the ICESat-2 observations are lower than 0.3 m, which is in range of the ICESat-2 uncertainties. The SOIPS also performs well on sea ice drift forecasts, both in magnitude and direction. Statistical analysis suggests that annual mean forecasting errors of sea ice drift at lead time of 168-
hour with respect to the NSIDC sea ice motion data are 2.22 cm/s in magnitude and 2.81° in direction. Furthermore, sea ice convergence rate, which can be derived from sea ice velocity forecasts, has a high potential in supporting ship navigation on local fine scale. A typical application of how sea ice convergence rate forecasts benefit for the icebreaker navigation in the Prydz Bay is illustrated. We realize that improvement on sea ice convergence rate forecasts may be achieved if we introduce a landfast ice parameterization into the SOIPS, which has been considered as one point of future model developments. Since
the involved ice-shelf model does not simulate collapse of ice-shelf (Ochwat et al., 2024) and the ice-shelf topography

remains unchanged in the SOIPS, replacing the simple static ice-shelf modular by a sophisticated thermodynamic–dynamic ice-shelf model may further improve the performance of the SOIPS on sea ice forecasts.

Satellite observations of sea ice concentration, thickness and drift have been used to estimate sea ice production and transport in the Antarctic coastal polynyas (Drucker et al., 2011; Nihashi et al., 2017; Tian et al., 2020). However, due to the relative scarce coverage of satellite observations in the Antarctic, especially in sea ice thickness, the evaluation of the SOIPS sea ice forecasts in this work still has considerable uncertainties. Part of the evaluation uncertainties come from the observational uncertainties themself, and part from the differences in spatial-temporal resolutions, as well as the coastline, between the SOIPS and the observations. Accurate short-term sea ice forecasts rely on optimized initial conditions at the forecasting onset, precise atmospheric forcing data (Pascual-Ahuir and Wang, 2023) if using an ice-ocean coupled model, and model physics in representing sea ice melt-freeze process and its heat and momentum exchanges with the underlying ocean. Specifically, the complex interactions among atmosphere, sea ice, ocean, ice-shelf, ice-sheet in the Antarctic region make the Antarctic sea ice forecasts more difficult. Moreover, in the Antarctic regional sea ice–ocean modeling, how to deal with oceanic open boundary conditions is a big challenge since the broad mid-latitude ocean surrounding the Antarctica can impact the Antarctic ocean and sea ice from all directions, i.e. Southern Pacific Ocean, Southern Atlantic Ocean, and Southern Indian Ocean. The utilizing of climatological monthly mean oceanic boundary conditions from the ECCO2 data makes the current configuration of the SOIPS lacking of interannual variance at the model boundary originating from ocean variability in lower latitudes. Although the Antarctic sea ice forecasts based on global models (Blockley et al., 2014; Posey et al., 2015; Smith et al., 2016; Lellouche et al., 2018; Johnson et al., 2019) carried out by international weather forecasting centers avoid the problem of dealing with oceanic boundary conditions, this newly developed regional sea ice forecasting system can operationally provide available sea ice forecasting information for the Southern Ocean at a moderate resolution and a high computational efficiency.

We have successfully applied synchronized assimilation of satellite-observed sea ice concentration, sea ice thickness, and sea surface temperature in our sea ice forecasting system for the Arctic, i.e. the Arctic Ice Ocean Prediction System (Mu et al., 2019; Liang et al., 2019). Owing to the rarity of operational satellite sea ice thickness observations with high spatial-temporal coverage in the Antarctic, the current version of the SOIPS only assimilates the AMSR2 sea ice concentration observations. In future, along with the elevation of satellite observation capacity, more and more sea ice and ocean variables are scheduled to be assimilated into the SOIPS to promote its ability on the Antarctic sea ice forecasts. Besides, more precise atmospheric forcing data, more advanced model sea ice-ocean physics, and more satellite and in situ observations are urgently needed to support the numerical sea ice forecasts for the Southern Ocean.

**Code and data availability.** The MODIS images is accessed at https://worldview.earthdata.nasa.gov; The WOA09 data is accessed at https://www.nodc.noaa.gov/OC5/WOA09; The GFS data is accessed at https://www.ncei.noaa.gov/products/weather-climate-models/global-forecast; The AMSR-E data is accessed at https://nsidc.org/data/AU_SI25/versions/1; The ICESat data is accessed at https://nsidc.org/data/nsidc-0304; The

ATLAS/ICESat-2 L3B data is accessed at https://nsidc.org/data/atl20/versions/4; The Polar Pathfinder data is accessed at https://nsidc.org/data/nsidc-0116/versions/4; The JRA-55 data is accessed at http://jra.kishou.go.jp/JRA-55; The AMSR2 data is accessed at https://seaice.uni-bremen.de/sea-ice-concentration; The OSISAF data is accessed at https://osi-saf.eumetsat.int/products/osi-401-d; The PDAF software is accessed at https://pdaf.awi.de/trac/wiki; The SOIPS used to produce the results in this paper can be accessed at https://zenodo.org/records/11381604.


**Author contributions.** F. Zhao conducted the SOIPS and data analysis, and composed the initial draft. X. Liang revised the initial draft. Z. Tian, M. Li, N. Liu and C. Liu contributed to data analysis and revising the initial draft.

**Competing interests.** The authors declare that they have no conflict of interest.


**Acknowledgments.** The authors sincerely thank the two anonymous reviewers for the constructive comments, and the topic editor Dr. Qiang Wang for the editorial work. The authors sincerely thank the NASA for the MODIS images; the NCEP for the WOA09 data and the GFS data; the NSIDC for the AMSR-E ice concentration data, the ICESat ice thickness data, the ATLAS/ICESat-2 L3B ice freeboard data, and the Polar Pathfinder ice motion data; the Japanese Meteorological Agency for
the JRA55 data; the University of Bremen for the AMSR2 ice concentration data; the Norwegian Meteorological Institute for the OSISAF ice concentration data. We also acknowledge the developer of the PDAF software.

**Financial support.** This work is supported by the National Key R&D Program of China (2022YFF0802000) and the National Natural Science Foundation of China (42276250).

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
