# Peer review of "Southern Ocean Ice Prediction System version 1.0 (SOIPS v1.0): description of the system and evaluation of synoptic scale sea ice forecasts"

_Geoscientific Model Development, 2024_

## Author Comment (AC1)

RC1:

General comments

In this study, the authors describe a new operational sea ice forecasting system for the Southern Ocean using a regional MITgcm ocean/sea ice/ice shelf general circulation model along with an ensemble based localized error Kalman Filter data assimilation system that assimilates sea ice concentration on a daily basis. Results from forecasts ranging from 24 to 168 hours are compared against different observational products to show the model performance in terms of RMSE of sea ice concentration, integrated ice-edge error, mean absolute error of ice thickness, and mean absolute error of sea ice drift. There was also a comparison of model sea ice convergence forecasts versus changes in MODIS imagery for one particular event (a sea ice opening in November 2021 that would be relevant for navigation to a particular coastal station).

I thought the manuscript was mostly (see below) clear and easy to understand. A regional sea ice forecast system for the Southern Ocean would certainly be useful, not only for the scientific/resupply missions for different nations, but also for the many private operations (i.e. fishing and tourism) that are becoming more numerous in Antarctic waters. MITgcm is a great tool for the ocean and ice shelf modeling. While the embedded sea ice model may not be the most "up to date", I think it is fine for these purposes, especially in the Antarctic where I do not think the lack of different ice thickness categories is such an issue where there is not much multi-year ice. I am not an expert on data assimilation and cannot comment on the appropriateness of the method used here (hopefully there will be a reviewer who can). All in all, this seems like a solid numerical setup for a sea ice forecast system (although I do have some questions below).

Response:

Dear reviewer, thanks a lot for your time and valuable comments on this manuscript. Our replies to your comments and suggestions are as follows.

My two concerns for a manuscript that is describing a forecast system is that I think more needs to be added to the model description and that it is hard to tell how well this setup is performing compared to either a simple forward model with no data assimilation or other existing global sea ice forecast models.

1) I think there are important aspects of the forward model that are relevant to dynamically moving ice around that are not described. Is there any parameterization of landfast ice processes? I know one exists in MITgcm, but that is meant more for the Arctic and it is not mentioned here, or in the Zhao et al. 2023 paper describing the forward model, whether it is used (and/or modified). Are icebergs (especially grounded ones that can limit ice transport) represented? Is tidal forcing included? All these processes would be relevant to ice motion and divergence.

Response:

Itkin et al. (2014) proposed a landfast ice parameterization for low salinity shallow shelf water in the Arctic marginal seas and tested its impacts on the stability of the Arctic halocline based on the MITgcm. Taking into account the unresolved shallow water topography and landfast ice internal strength, the parameterization sets the maximum compressive strength of landfast ice to the double of the drift ice inside the prescribed maximal landfast ice edge mark. Liu et al. (2022) proposed a more complicated landfast ice parameterization that uses lateral drag as a function of sea ice thickness, drift velocity, and local coastline length. Their simulation suggested that the parameterization leads to an improved and realistic landfast ice distribution in most marginal seas in the Arctic.

Our model does not include landfast ice parameterization. Although without landfast ice parameterization, the model still has a capacity to simulate the nearly immobile sea ice zone attached to the coast of the Antarctica, which can be inferred from Figure 10 in the original manuscript. The capacity originates from the correct simulation of sea ice thickness and drift velocity in the landfast ice zone.

Iceberg parameterization and tide forcing are also not applied in our model. We admit that all these processes participate in regulating sea ice motion and divergence, and we thank you for pointing out these as our future model developing direction.

We have clarified relevant model settings. The statement of "Neither specific landfast ice parameterization, iceberg parameterization, nor tide forcing has been involved in the SOIPS." has been added into the revised manuscript.

References:

Itkin, P., M. Losch, and R. Gerdes (2015), Landfast ice affects the stability of the Arctic halocline: Evidence from a numerical model, J. Geophys. Res. Oceans., 120, 2622–2635, doi:10.1002/2014JC010353.

Liu, Y., M. Losch, N. Hutter, and L. Mu (2022), A new parameterization of coastal drag to simulate landfast ice in deep marginal seas in the Arctic, J. Geophys. Res. Oceans., 127, e2022JC018413, doi: 10.1029/2022JC018413.

2) The manuscript has several descriptions of the performance of the data assimilative forecasts, but it is difficult to tell how much the data assimilation adds to the forecast skill. Was there a control run (like in Liang et al., 2019, JGR, which has some of the same authors) without data assimilation over the same dates as the forecast runs? If so, what does that look like compared to observations? What do the ice concentration and integrated ice-edge errors look like over time with no forward modeling and just the initial analysis (i.e. persistence of the initial sea ice state throughout the forecast period)? I do not expect the authors to go through all the different existing global sea ice forecasting systems, but I think it might be helpful to a reader if some information were given on how this system compares to others. For example, the 168-hour sea ice concentration RMSE for this model (Figure 3) looks considerably better for most months than the GIOPS for the Southern hemisphere (Figure 3b in Smith et al., 2016).

Response:

Thanks a lot for this comment. We renamed the original experiment as DA_Forecast in the revised manuscript. According to your suggestions, we have conducted a control run without any data assimilation (NoDA_Forecast) and also performed persistence forecast (PE_Forecast) in the analysis. We believe that these additional analysis can substantially improve the quality of our manuscript.

Regarding to sea ice concentration forecast (Figure R1), the DA_Forecast run performed best and the NoDA_Forecast run performed worst in most time except during late March–early June. Since the PE_Forecast run uses the initial condition of the DA_Forecast run which assimilates sea ice concentration observations as forecasts of the following 168 hours, the PE_Forecast run generally performed better than the NoDA_Forecast run and worse than the DA_Forecast run. During late March–early June, the PE_Forecast run performed worse than the other two runs at lead time of 168-hour, suggesting that the sea ice changes rapidly in response to the oceanic and atmospheric forcing during this onset–to–fast freezing period.

Regarding to the IIEE forecast (Figure R2), the DA_Forecast run performed best and the NoDA_Forecast run performed worst over the whole study period.

Regarding to sea ice thickness forecast (Figure R3), the DA_Forecast run also performed better than the NoDA_Forecast run at lead time of 24-hour. Sea ice thickness changes relatively small in one day, and sea ice thickness forecast of the PE_Forecast run is approximate equal to that of the DA_Forecast run at lead time of 24-hour, so the PE_Forecast run is not marked on this figure.

We also compare our forecasts to the physical analysis field of the Antarctic Ocean produced by Mercator Ocean International (MOI; accessed at https://data.marine.copernicus.eu/product/GLOBAL_ANALYSISFORECAST_PHY_ 001_024/description). The MOI product is physical analysis data and free accessed. We can not download the GIOPS data from the internet.

With respect to the OSISAF data, the RMSE of sea ice concentration forecasts of the DA_Forecast run at lead time of 24-hour is larger than that of the MOI product (Figure R4), while the IIEE of the DA_Forecast run at lead time of 24-hour is close to that of the MOI product (Figure R5). Note that the MOI product has assimilated the OSISAF sea ice concentration data, the SOIPS assimilated the AMSR2 sea ice concentration data, thus the MOI product has a lower RMSE of sea ice concentration when uses the OSISAF data as validation reference in this study.

Detailed comparison among the different runs and validation of our system against the observations (Figure R1, R2, R3) are presented in the revised manuscript. We put Figure R4, R5 into the supplementary material.

[Figure]

*Figure R1. Time series of the RMSEs of the assimilated AMSR2 data and sea ice concentration forecasts at different lead time with respect to the OSISAF data. The blue, green, yellow, red, and black solid lines denote the sea ice concentration forecasts of the DA_Forecast run at lead time of 24-hour, 72-hour, 120-hour, 168-hour, and the AMSR2 data, respectively. The blue and red long-dashed lines denote the sea ice concentration forecasts of the NoDA_Forecast run at lead time of 24-hour and 168-hour, respectively. The blue and red short-dashed lines denote the sea ice concentration forecasts of the PE_Forecast run at lead time of 24-hour and 168-hour, respectively.*

[Figure]

*Figure R2. Time series of the IIEE of the assimilated AMSR2 data and the forecasts at different lead time with respect to the OSISAF data. The blue, green, yellow, red, and black solid lines denote the IIEE forecasts of the DA_Forecast run at lead time of 24-hour, 72-hour, 120-hour, 168-hour, and the AMSR2 data, respectively. The blue and red long-dashed lines denote the IIEE forecasts of the NoDA_Forecast run at lead time of 24-hour and 168-hour, respectively. The blue and red short-dashed lines denote the IIEE forecasts of the PE_Forecast run at lead time of 24-hour and 168-hour, respectively.*

[Figure]

*Figure R3. Time series of the mean sea ice thickness and uncertainties of the ICESat-2 observations (black and green lines), the sea ice thickness forecasts at lead time of 24-hour of the DA_Forecast and NoDA_Forecast runs (red and blue solid lines), and the mean absolute errors between the forecasts and observations (red and blue dashed lines).*

[Figure]

*Figure R4. Time series of the RMSEs of the assimilated AMSR2 data, the MOI product, and sea ice concentration forecasts of the DA_Forecast run at different lead time with respect to the OSISAF data. The blue, green, yellow, red, black, and purple lines denote the sea ice concentration forecasts at lead time of 24-hour, 72-hour, 120-hour, 168-hour, the AMSR2 data, and the MOI product, respectively.*

[Figure]

*Figure R5. Time series of the IIEE of the assimilated AMSR2 data, the MOI product, and the forecasts of the DA_Forecast run at different lead time with respect to the OSISAF data. The blue, green, yellow, red, black, and purple lines denote the forecasts of the DA_Forecast run at lead time of 24-hour, 72-hour, 120-hour, 168-hour, the AMSR2 data, and the MOI product, respectively.*

There are also several minor awkward usages or subject/verb agreement mistakes.  I am not going to explicitly comment on most of them, and they generally do not make

the manuscript more difficult to understand (I still feel the manuscript is pretty well organized and understandable), but I think they should be cleared up in the next version.

I have some other specific comments and suggestions below, but most of these are minor and should be easily dealt with by the authors. I think SOIPS may be a very good forecast system for sea ice, but this manuscript still needs some work before it can help an interested reader judge that for themselves.

Specific comments

Line 26: By "with thin first-year ice dominating the majority" do the authors mean that the majority of the ice is thin first-year ice?

Response:

Yes. We revised this sentence to "This situation is partly caused by the natural feature of Antarctic sea ice that the majority of the ice is thin first-year ice."

Line 28: Since katabatic winds can blow sea ice away from the coast, as well as away from the front of ice shelves, suggest changing "off the ice-shelf" to "off the ice-shelf and coast".

Response:

Revised.

Lines 61-63: I agree with the authors that regional models "with higher resolution" still offer significant advantages, but isn't the resolution of this model (line 93: ~ 18 km) lower than the resolution at these latitudes of most of the global models (1/4 degree or better) listed in this paragraph?

Response:

I agree. We revised this sentence to "Although resolution of global models is constantly becoming finer, regional ice–ocean coupled models with lower computational cost still offer some advantages when appropriate initial and boundary conditions are adopted".

Line 92: I think it is also worth mentioning that the open boundaries are farther north than any likely northern extent of the sea ice.

Response:

We revised this sentence to "The ocean model uses curvilinear coordinates with the open boundaries far north away from the domain of the Antarctic Circumpolar Current (ACC) and any likely northern extent of the sea ice.".

Line 95: Large and Pond (1981) is just the bulk formula for momentum flux (I think). Is there a bulk formulation used for heat and salt/freshwater fluxes?

Response:

We revised this sentence to "The ocean model utilizing the finite-volume incompressible Navier-Stokes equations adopts the bulk formula for heat and momentum calculation at surface (Large and Pond, 1981, 1982)".

Line 99: Suggest adding the Losch 2008 reference that describes the implementation of ice shelves in MITgcm.

Response:

Added.

Line 107 (and line 376): I do not think Zwally, 1990 is the best reference for the initial ice thickness data and the URL given on line 376 is the ICESat 500m DEM, not the sea ice thickness. Is this the Kurtz and Markus (JGR, 2012) data?

Response:

Yes. We revised the reference to Kurtz and Markus (2012).

Line 128: Were any experiments done with more or less ensemble members?

Response:

NO. We have not carried out any experiment to test the impact of ensemble size on forecasting ability for the Antarctic system. The choice of 12 ensemble members is made according to the setting of our Arctic system and the limitation of computational resource of operational implementation.

Lines 129-132: I assume the ensemble is generated in the method described in the PDAF wiki (https://pdaf.awi.de/trac/wiki/EnsembleGeneration), but there is no reference, and very few details, on how it is generated.

Response:

The ensemble is generated using the method described in the PDAF wiki. We added "(Pham, 2001)" into the revised manuscript.

Lines 174-175 and Figure 3: I agree that the bias between AMSR2 and OSISAF partially explains the ice concentration forecasting errors, but the shorter term forecasts (24 and 72 hours) look to generally be better than the assimilated AMSR2 data. Do the authors have any explanation for this? What would the forecast errors be with no data assimilation?

Response:

In the DA_Forecast run, the AMSR2 data was assimilated into the ensemble of model restart fields, and an analyzed (updated) ensemble of model restart fields was generated. Initialized from the analyzed ensemble of model restart fields, each ensemble member was integrated for 168 hours driven by atmospheric forcing. So the

24-hour and 72-hour forecasts included not only the observational information, but also sea ice changes generated by model physics. The NoDA_Forecast run performed worse than the DA_Forecast run in most time (Figure R1; also see the response to your major comment 2).

Figure 4: I think it would be helpful to have a figure like this (monthly RMSE for ice concentration) for the AMSR2 vs. OSISAF for comparison. I can certainly understand the authors not wanting to add any figures to the primary manuscript, but perhaps in a supplementary material section?

Response:

The monthly patterns of the RMSEs of sea ice concentration between the AMSR2 and OSISAF data (Figure R6) generally resemble and set the base for those between the 24-hour forecasts and the OSISAF data. We put Figure R6 into the supplementary material.

[Figure]

*Figure R6. Monthly patterns of the RMSEs of sea ice concentration between the AMSR2 and OSISAF data. (a)–(l) denote October 2021–September 2022.*

 As mentioned above, what about other non-simulated (I think, it is never explicitly stated one way or another) barriers to sea ice free drift such as fast ice or grounded icebergs?

Response:

Thanks for the suggestion. The statement of "The lack of specific landfast ice parameterization may lead to unrealistic landfast ice zones around the Antarctica, which possibly also contribute to the mismatch of sea ice edges." has been added into the revised manuscript.

Figure 6: The sea ice edge forecasts look really good at 24-hours. If the authors do create a supplementary material section, I would be curious what the ice edge looks like at 168-hour lead time.

Response:

Comparing to the sea ice edge forecasts at lead time of 24-hour, the biases of sea ice edge forecasts at lead time of 168-hour with respect to the OSISAF data (Figure R7) are larger in November–December, March–April, and July–August. The areas with obvious bias amplification locate in southeastern Atlantic Ocean sector, southwestern Indian Ocean sector, and the southwestern Pacific Ocean sector. We put Figure R7 into the supplementary material.

[Figure]

*Figure R7. Monthly patterns of sea ice edge forecasts at lead time of 168-hour with respect to the OSISAF data. (a)-(l) denote October 2021−September 2022. The blue lines denote the DA_Forecast run. The red lines denote the OSISAF data. The gold contours denote the IIEE.*

Figure 8: As for figure 6, what does the ice thickness look like at 168-hour lead time?

Response:

The biases of sea ice thickness forecasts at lead time of 168-hour with respect to the ICESat2 data (Figure R8) do not change obviously in comparison with those of 24-hour. We put Figure R8 into the supplementary material.

[Figure]

*Figure R8. Seasonal patterns of the Antarctic sea ice thickness. The columns from left to right denote the DA_Forecast run at lead time of 168-hour, the ICESat2 observations, their deviations, and the uncertainties of the ICESat2 observations, respectively. The panels from top to bottom denote October–December, January–March, April–June, and July–September, respectively.*

Lines 279-280 and Figure 9b: I agree that it is mostly true that "the MAEs of both magnitude and direction of the sea ice drift forecasts do not exhibit significant amplification", but the MAE of the drift angle does increase significantly at 168 hours (compared to shorter lead times) in Oct-Nov and Jun-Sep.

Response:

We revised the sentence to "Along with the prolong of the forecast lead time, the MAEs of the sea ice drift magnitude forecasts do not exhibit significant amplification, and those of direction forecasts grow significantly at lead time of 168-hour in October–November and June–September.".

Lines 288-289: The mean absolute errors and mean magnitude of the NSIDC drift velocities are given earlier, but I did not see anything to indicate whether the mean model bias with respect to NSIDC was positive or negative until here. Apologies if I missed it, but is the mean difference (not mean absolute error) or mean drift velocity from the model given anywhere?

Response:

Thanks for the comment. We do not mention whether the mean model bias with respect to NSIDC is positive or negative in the original manuscript. The biases of sea ice drift magnitude between the DA_Forecast run at lead time of 24-hour and the NSIDC data are not uniform (Figure R9). In general, the DA_Forecast run produces larger magnitude of sea ice drift in the northern marginal sea ice zone and the coastal areas, while in between the DA_Forecast run produces smaller magnitude of sea ice drift.

The statement of "In general, the DA_Forecast run produces larger magnitude of sea ice drift in the northern marginal sea ice zone and the coastal areas, while in between the DA_Forecast run produces smaller magnitude of sea ice drift." has been added into the revised manuscript. We also put Figure R9 into the revised manuscript.

[Figure]

*Figure R9. Monthly patterns of the sea ice drift magnitude biases between the DA_Forecast run at lead time of 24-hour and the NSIDC sea ice drift data. (a)-(l) denote October 2021−September 2022.*

Line 298: Since landfast ice also floats, I suggest changing "Floating sea ice" to "Drifting sea ice" or "Moving sea ice".

Response:

We revised "Floating sea ice" to "Drifting sea ice".

Line 299: Same comment as above about "floating sea ice zone".

Response:

Revised.

Figure 10: What is the lead time for the forecasts on the left?   Also, Figures c and d are really impressive!

Response:

Thanks for this comment. This typical case was derived from the operational implementation of the SOIPS forecasts. The forecast was initialized on November 18, 2021, and Figure 10a, 10c, 10e denote the 24-hour, 48-hour, 72-hour forecasts.

We added the statement of "The forecast was initialized on 2021 November 18." into the figure caption.

Lines 337-338: Same point as for Lines 218-221 above.

Response:

We revised the sentence to "It should be mentioned that mismatch of sea ice edges in some nearshore areas originates from the divergence of coastlines, ice-shelf fronts or unrealistic landfast ice zones in the model domain and the OSISAF data.".

Lines 342-343:   Do the authors have any thoughts on if the forecasted convergence rates near the coast would be improved if the model included fastice processes?

Response:

Thanks for your suggestion. The statement of "Meanwhile, we realize that improvements on sea ice convergence rate forecasts may be achieved if we introduce a landfast ice parameterization into the SOIPS, which has been considered as one of future directions of model development." has been added into the revised manuscript.

Lines 358-363: I still think more needs to be done to show if this model can do a better (or at least similar) job compared to those other forecast systems.

Response:

We briefly compare our forecasts to the MOI product, and this part is presented in the supplementary material.

Lines 380-381: The zenodo link to SOIPS (https://doi.org/10.5281/zenodo.10457661) did not work.

Response:

The link has been updated to https://zenodo.org/records/11381604.

Technical corrections

Again, this is not a complete list and there are many minor grammatical errors that should be cleaned up in the next version.

Abstract lines 18, 19, and 20 and many other places:  Suggest changing "leading time" to "lead time".

Response:

All revised.

Line 36:  "Grahams Land" should be "Graham Land".

Response:

Revised.

Line 83:  Suggest changing "promise capacity" to "promise" or "capacity".

Response:

Revised to "capacity".

Line 124:  Should "5-order" be "5th-order"?

Response:

Revised to "5th-order".

Line 195:  Suggest changing "just a number of sea ice extent" to "just a sea ice extent number".

Response:

Revised.

Line 277:  Suggest changing "In contrary" to "In contrast".

Response:

Revised.

---

## Author Comment (AC2)

RC2:

In this study, Zhao et al. present an operational Southern Ocean Ice Prediction System and exhibit its ability for Antarctic sea ice prediction on synoptic time scales. They developed the prediction system based on MITgcm and assimilated satellite-derived sea ice concentration data, making predictions for the future 7 days. The prediction system shows promising skill in predicting the sea ice concentration, sea ice thickness, sea ice drift, and sea ice convergence.

Considering the limited effort for the operational Antarctic sea ice prediction when compared to its Arctic counterpart, this study is valuable by providing evidence of the model's ability for skillful Southern Ocean and sea ice prediction. In addition, the manuscript is well-organized and easy to understand. However, I found some points to be further clarified, which are listed below. I suggest a major revision is needed.

Response:

Dear reviewer, thanks a lot for your time and valuable comments on this manuscript. In the revised manuscript, we rename the original experiment as DA_Forecast run, and involve two additional experiments in the analysis: a experiment without any data assimilation (NoDA_Forecast) and a experiment of persistence forecast (PE_Forecast). The setting of the NoDA_Forecast run is the same to the DA_Forecast run except that no observational data has been assimilated. The PE_Forecast run uses the initial condition of the DA_Forecast run as forecasts of the following 168 hours. Note that the PE_Forecast run includes the observational sea ice concentration information due to data assimilation. Our replies to your comments and suggestions are as follows.

Major comment:

1. Despite the main point of this work being to demonstrate the ability of the prediction system for the operational Antarctic sea ice prediction, the added scientific

discussions will improve the manuscript a lot. The following are a few examples, but not limited to these.

(1)Why is the RMSE of prediction in Fig. 3 smaller than the RMSE of observation February and March? Why does the RMSE of prediction peak in April?

Response:

The AMSR2 data was assimilated into the ensemble of model restart fields on a daily basis, and an analyzed (updated) ensemble of model restart fields was generated. The analyzed model restart fields combined the modeled sea ice states with the observational sea ice states. Initialized from the analyzed ensemble of model restart fields, each ensemble member was integrated for 168 hours driven by atmospheric forcing. So the forecasts included not only the observational information, but also sea ice changes generated by model physics, which caused the better performance of the DA_Forecast run in comparison with that of the AMSR2 data, especially at lead time of 24-hour and 72-hour in January–early March and May–September.

Figure 4 shows that large sea ice concentration RMSE appears in most areas of sea ice zone around the Antarctica in March–April, suggesting that the model has a relative low capacity in correctly simulating sea ice growth rate during this onset–to–fast freezing period. This partly originates from that the sea ice model in the SOIPS uses the zero-layer ice/snow thermodynamics (Semtner, 1976), which is a simple sea ice model compared to sophisticated multi-layer ice/snow thermodynamical models.

We added the statement of "The AMSR2 sea ice concentration data was assimilated into the ensemble of model restart fields on a daily basis, and an analyzed (updated) ensemble of model restart fields combining the modeled and observational sea ice states was generated, which were further integrated for 168 hours driven by atmospheric forcing. The forecasts included not only the observational information, but also sea ice changes generated by model physics. This causes the better performance of sea ice concentration forecasts in the DA_Forecast run in comparison

with that of the AMSR2 data, especially at lead time of 24-hour and 72-hour in January–early March and May–September. On the other side, large sea ice concentration RMSE appears in most areas of sea ice zone around the Antarctica in March–April, suggesting that the model has a relative low capacity in correctly simulating sea ice growth rate during this onset–to–fast freezing period. This probably originates from that the sea ice model in the SOIPS uses the zero-layer ice/snow thermodynamics, which is a simple sea ice model compared to sophisticated multi-layer ice/snow thermodynamical models." into the revised manuscript.

(2)L180-190: it's interesting to know how many errors can be explained by the difference between OSISAF and AMSR2 and how many are caused by error growth during the model integration.

Response:

The monthly patterns of the RMSEs of sea ice concentration between the AMSR2 and OSISAF data (Figure R1) show large values in the northern marginal ice zone and the coast while small values in between, which sets the base for those between the forecasts and the OSISAF data. Due to the large spatial-temporal differences of the sea ice concentration RMSE, it is hard to quantitatively clarify how many errors are caused by error growth during the model integration. As a reference, with respect to the OSISAF data, the annual mean RMSEs of the AMSR2 data, the forecasts at lead times of 24-hour, 72-hour, 120-hour and 168-hour are 0.165, 0.15, 0.16, 0.17 and 0.19, respectively. The rates of the RMSE of the forecasts to the AMSR2 data are 91%, 97%, 103%, and 115%, respectively. We put Figure R1 into the supplementary material.

[Figure]

*Figure R1. Monthly patterns of the RMSEs of sea ice concentration between the AMSR2 and OSISAF data. (a)−(l) denote October 2021−September 2022.*

(3) What model deficiency in Fig. 5 leads to an increase in predicted IIEE in March-April and a decrease in April-May? Why is there little difference in IIEE for different lead times in January-June, but significant differences in other months?

Response:

As mentioned in the response to your major comment 1(1), the model has a relative low capacity in correctly simulating sea ice growth (expansion) rate during March–April (the onset–to–fast freezing period). This probably originates from that the sea ice model in the SOIPS uses a simple zero-layer ice/snow thermodynamics.

Figure R2 shows the monthly patterns of sea ice edge forecasts at lead time of 168-hour with respect to the OSISAF data. In comparison with July–December, the sea ice zone is smaller during January–June, so the integrated ice-edge error grows moderately in response to prolonged forecast lead time. Moreover, the sea ice edge locates more north during July–December, and the marginal ice zone is more close to the ACC-impacting areas where active oceanic and atmospheric dynamical processes promote the amplification of the integrated ice-edge error along with the prolong of forecast lead time.

We added the statement of "In comparison with July–December, the sea ice zone is smaller during January–June, so the IIEE grows moderately in response to prolonged forecast lead time. Moreover, the sea ice edge locates more north during July–December, and the marginal ice zone is more close to the ACC-impacting areas where active oceanic and atmospheric dynamical processes promote the amplification of the IIEE along with the prolong of forecast lead time." into the revised manuscript.

We put Figure R2 into the supplementary material.

[Figure]

*Figure R2. Monthly patterns of sea ice edge forecasts at lead time of 168-hour with respect to the OSISAF data. (a)-(l) denote October 2021−September 2022. The blue lines denote the DA_Forecast run. The red lines denote the OSISAF data. The gold contours denote the IIEE.*

(4)In Fig. 9, why the evolution of forecast errors in magnitude of sea ice drift is different from that in direction? Additionally, due to the complexity of the South Pacific Ocean current system, it is recommended to showcase the drift forecast capability in more ways, such as its spatial distribution.

Response:

Similar to the patterns of sea ice concentration RMSE (Figure 4 in the original manuscript), the monthly patterns of the magnitude bias between the sea ice drift forecasts at lead time of 24-hour and the NSIDC data (Figure R3) show large values in the northern marginal ice zone and the coast, while small values in between. During January–March, the Antarctic sea ice zone shrinks to its annual minima, large biases in magnitude of sea ice drift occur in most sea ice areas, and thus the mean absolute error in magnitude of sea ice drift forecasts is large during January–March (Figure 9 in the original manuscript). In other months, large biases in sea ice drift direction forecasts also occur in the densely packed sea ice zone, especially the Bellingshausen-Amundsen-Ross Seas and the southeastern Antarctic Ocean sector (Figure R4), thus the mean absolute error in direction of sea ice drift forecasts is large in other months. We added Figure R3, R4 into the revised manuscript.

[Figure]

*Figure R3. Monthly patterns of the magnitude bias between the sea ice drift forecasts at lead time of 24-hour and the NSIDC data. (a)-(l) denote October 2021–September 2022.*

[Figure]

*Figure R4. Monthly patterns of the absolute bias between the direction of sea ice drift forecasts at lead time of 24-hour and the NSIDC data. (a)-(l) denote October 2021– September 2022.*

2. Compared to other studies, an important feature of this research is the incorporation of ice-shelf model. Thus,

(1) Please provide more details on the ice-shelf model and coupling method. According to Line 98-99, it's hard to realize the differences between the ice-shelf model used here and boundary conditions used in previous studies.

Response:

We apologize for the misleading statement. At the current stage, the ice-shelf modular in the MITgcm is not a sophisticated ice-shelf model, yet this ice-shelf model can still function as an effective static boundary condition. This ice-shelf model was developed by Losch (2008). Since Losch (2008) has provided a description of this ice-shelf model in detail, we have not repeated the documentation of this model in this study. We agree with the reviewer that we should describe the ice-shelf model more clearly.

The ice-shelf model affects the coupled model system through dynamics and thermodynamics. Dynamically, the ice shelf draft on the top of the water column has a similar role as the surface orography. Underneath an ice shelf, the pressure at the top of the water column is the sum of the atmospheric pressure and the weight of the ice shelf column. Thermodynamically, the freezing and melting at the basal surface of the ice shelf can induce effective heat flux and virtual salt flux at the ice–ocean interface, with an additional tendency term of temperature and salinity to the ocean at the depth of the ice-shelf draft. Then, a boundary layer between the ice shelf and the ocean is formed. In addition, the application of partial cells has also been introduced in the ice-shelf model, and thereby it can properly represent the geometry of the sub-ice-shelf cavity and allow for an accurate and smooth solution at the ocean–ice-shelf interface.

We added the statement of "The ice-shelf, serving as as a static surface boundary condition, exerts dynamic and thermodynamic influences on the underlying ocean and

thus affects ocean circulation and sea ice (Losch, 2008). Dynamically, the ice shelf draft on the top of the water column has a similar role as the surface orography. Underneath an ice shelf, the pressure at the top of the water column is the sum of the atmospheric pressure and the weight of the ice shelf column. Thermodynamically, the freezing and melting at the basal surface of the ice shelf can induce effective heat flux and virtual salt flux at the ice–ocean interface, with an additional tendency term of temperature and salinity to the ocean at the depth of the ice-shelf draft. Then, a boundary layer between the ice shelf and the ocean is formed. In addition, the application of partial cells has also been introduced in the ice-shelf model, and thereby it can properly represent the geometry of the sub-ice-shelf cavity and allow for an accurate and smooth solution at the ocean–ice-shelf interface." into the revised manuscript.

(2) More analyses should be conducted to highlight the advantages of this feature. For example, the Larsen-B ice shelf collapsed in January 2022 (doi: 10.5194/tc-2023-88), which occurred during the experimental period, so it is advisable to investigate the impact of this event on sea ice assimilation and prediction.

Response:

Since the ice-shelf model functions as a static surface boundary condition, the ice-shelf model does not simulate collapse of ice-shelf, and the ice-shelf topography remains unchanged during the experimental period.

We added the statement of "On the eastern side of the Antarctic Peninsula, the multi-year landfast ice in the northern Larsen B embayment breakout and disentangled from the Larsen B ice shelf in January 2022 (Ochwat et al., 2024). Since the involved ice-shelf model does not simulate collapse of ice-shelf and the ice-shelf topography remains unchanged in the SOIPS, replacing the simple static ice-shelf modular by a sophisticated thermodynamic–dynamic ice-shelf model may further

improve the performance of the SOIPS on sea ice forecasts." into the revised manuscript.

Minor comment:

Line 66-71: Because the preceding paragraph mentioned the advantages of regional models, it might be better to illustrate data assimilation studies based on regional models, such as SOSE.

Response:

We added the statement of "The Southern Ocean State Estimate (Mazloff et al., 2010) constrains model state using in situ and satellite measurements through 4D-Var data assimilation." into the revised manuscript.

Line 82: Considering the submission is in 2024 and an operational forecasting system is involved, the experiment should be extended to include 2023 when the Antarctic sea ice reaches its minimum extent.

Response:

Thanks for the suggestion. We prefer to keep the original study period in the revised manuscript. Meanwhile, we have validated the sea ice extent forecasts before September 2023 in the operational record and put Figure R5 into the supplementary material. The minimum sea ice extent forecasts of the DA_Forecast run at lead time of 24-hour are $1.73 \times 10^6$ $km^2$ in 2022 and $1.49 \times 10^6$ $km^2$ in 2023. The minimum sea ice extent derived from the AMSR2 data are $1.76 \times 10^6$ $km^2$ in 2022 and $1.63 \times 10^6$ $km^2$ in 2023. The SOIPS predicted a lower sea ice extent minimum in 2023 than in 2022.

[Figure]

*Figure R5. Sea ice extent evolution of the AMSR2 data (black line) and the DA_Forecast run at lead time of 24-hour (blue line), 72-hour (green line), 120-hour (yellow line), and 168-hour (red line).*

Line 93: Considering that one important application of this system is for shipping services, the higher model resolution would indeed be preferable. Therefore, why not use a higher-resolution model such as MITgcm with 1/6° (doi: 10.1002/2016jc012650)?

Response:

We agree with your comment. At current stage, the use of low-resolution MITgcm model in the SOIPS is determined by the limitation of computational resource in the operational implementation. We have cited Verdy and Mazloff (2017) in the revised manuscript.

Line 130-132: Please provide more details on the initial field perturbation process, such as which variables are perturbed? What is the explained variance of the first 11 EOF modes?

Response:

We revised the sentence to "The initial ensemble of SOIPS is generated by disturbing the latest state of the model free run including sea ice concentration and thickness.".

We have cited Pham (2001) in the revised manuscript, which introduces the method of applying an order-2 sampling scheme to leading EOF modes to generate perturbation.

The explained variances of the first and the 11th EOF modes are 47.48% and 0.66%, respectively. The first 11 EOF modes account in total for 69.34% of total variance.

Line 135-136: Please provide more information about the observational errors used in the assimilation. For example, is 0.15 the representative error of observations? If so, how are instrument errors identified?

Response:

We used a uniform value of 15% as the representative error of the AMSR2 sea ice concentration observations for simplicity in the SOIPS. We don't know how the instruction errors are identified, but according to the manual of the AMSR2 sea ice concentration product, the AMSR2 observations have different errors in different sea ice concentration ranges. In densely packed sea ice zone, the instrument error should be lower than 15%.

Line 138-140: The author's previous study used JRA55 as the atmospheric forcing, while this study uses GFS. Given the importance of atmospheric forcing for Antarctic sea ice simulation, did the author optimize the model parameters after changing the atmospheric forcing, as in doi: 10.1016/j.ocemod.2023.102183? If optimization has been conducted, are there significant changes in the model parameters? If not, could some of the subsequent results be attributed to the mismatch between the atmospheric forcing and the model, such as Line 213-214?

Response:

The JRA55 data is reanalysis data which can not be used to drive operational sea ice forecasts. The GFS product is an operational weather forecasting product.

We did not optimize the model parameters. According to our experience of polar sea ice modeling, the zero-layer ice/snow thermodynamics have low capacity in correctly simulating sea ice extent expand/shrink rate during melt/freeze transition period. We suspect that the mismatch between forecasts and observations in March–April originates from use of the zero-layer ice/snow thermodynamics, rather than from the change of atmospheric forcing.

We have cited Pascual-Ahuir and Wang (2023) in the revised manuscript.

Line 155: is it OSI-401-d?

Response:

The data ID is OSI-401-b before 24 April 2023, thereafter changed to OSI-401-d.

We have updated the data statement in Code and data availability.

Line 163-165: I would argue that the RMSE increases to the end of March, followed by a decrease starting from April.

Response:

We revised the sentence to "Basically the RMSEs between the SOIPS forecasts and OSISAF data gradually increase during October–March (hereafter the latter month in such expressions that the latter month is earlier than the former month denotes the month of the next year) followed by a decrease starting from April.".

Line 208-209: It's hard to follow and please rewrite this sentence.

Response:

We revised the sentence to "With respect to the OSISAF data, the curves of IIEEs of the DA_Forecast run at different lead times have similar shapes to that of the assimilated AMSR2 data.".

Line 219-221: It's very interesting and It would be more valuable if the author could present the correction method and the corrected IIEE.

Response:

Thanks for the comment. We will perform the IIEE correction in future work.

Line 252-253: It's recommended to add the uncertainty of ICESat-2 to Fig. 8. From Fig. 7, the uncertainty appears to be around 0.5m, while in Fig. 8, the prediction error in the southern Weddell Sea and the western Ross Sea seem to reach up to 0.6m. Are these errors beyond the uncertainties of the observation? Why are the prediction errors of SIT larger in these areas?

Response:

We have added the ICESat-2 uncertainty into the figure and replaced the original Figure 8 by Figure R6 in the revised manuscript. The prediction errors in the southern Weddell Sea are in the range of the ICESat-2 uncertainty, but the prediction errors in the western Ross Sea are out of the range of the ICESat-2 uncertainty. We suspect that the larger SIT bias in these areas are caused by the poor simulation of growth rate of sea ice thickness during the freezing seasons, partly originating from the biases in the simulated ocean temperature or air temperature in the GFS data.

[Figure]

*Figure R6. Seasonal patterns of the Antarctic sea ice thickness. The columns from left to right denote the DA_Forecast run at lead time of 24-hour, the ICESat2 observations, their deviations, and the uncertainties of the ICESat2 observations, respectively. The panels from top to bottom denote October–December, January–March, April–June, and July–September, respectively.*

Line 295: Please provide the specific definition of Sea ice convergence rate. What are the similarities and differences between the sea ice convergence rate and the divergence of sea ice drift?

Response:

We defined sea ice convergence rate (SICR) as $SICR = -(\partial u_m / \partial x + \partial v_m / \partial y)$ (negative value represents sea ice dispersion, positive value represents sea ice accumulation). $(u_m, v_m)$ are the ice drift components on the model coordinates. Sea ice convergence rate is the opposite of the divergence of sea ice drift.

We revised the sentence to "Sea ice convergence rate (SICR), defined as $SICR = -(\partial u_m / \partial x + \partial v_m / \partial y)$ (negative value represents sea ice dispersion, positive value represents sea ice accumulation), is a useful metric in guiding ship navigation in sea ice zone.".

There are quite a few typos. For instance, an extra hyphen of "synoptic-scale" in Line 332 and an extra left parenthesis in Line 359.

Response:

All revised.

---

## Author Response (AR2)

General comments

I thank the authors for all their efforts in response to my previous comments, especially the new control run with no data assimilation and the new persistence forecast. I know it was a fair bit of work, but I do think those two runs will be quite helpful in enabling the audience to judge the utility of the data assimilation in this forecast system.

I still think a little more could be done to compare this system to other ice forecasts systems. The additional comparisons to the MOI product in the Supplement are great, but there is no mention in the main text of what these show (just a statement that comparisons are made in the Supplement). I think a simple sentence or two of what Figures S4 and S5 show could be added to the main text. With respect to GIOPS, I understand that the data could not be downloaded, but I wasn't expecting the authors to download the data and make new figures. Smith et al. (2016) already include some of the same comparisons for the Southern Hemisphere as done here (sea ice concentration RMSE as a function of lead time, 168 hr lead time ice concentration RMSE as a function of time of year) that could just be mentioned in the this text.

There still are several minor awkward usages or subject/verb agreement mistakes. Again, I am not going to explicitly comment on most of them, and they still do not make the manuscript more difficult to understand, but I really think some effort should be put into cleaning them up.

I have a few new specific comments and suggestions below, but all of these are minor and should be easily dealt with by the authors. I still think some minor revision is necessary before publication, but I also think this is close to being a good guide for readers into how well SOIPS works (pretty well in my estimation) as a forecast system for sea ice in the Southern Ocean.

Response:
Dear reviewer, thanks a lot for your time and valuable comments on this manuscript. We have added several sentences into the main text to show the comparison among the SOIPS forecasts, the MOI product and the GIOPS forecasts.

"Additionally as a reference, the sea ice concentration RMSE of the GOIPS forecasts at lead time of 168-hour maintains below 0.35 in the year of 2011 with respect to the Interactive Multisensor Snow and Ice Mapping System ice extent product (Helfrich et al., 2007). With respect to the OSISAF data, the sea ice concentration RMSE of the SOIPS forecasts at lead time of 24-hour is larger than that of the MOI product. It should be mentioned that the MOI product has assimilated the OSISAF sea ice concentration data, which leads to a lower RMSE in comparison to the SOIPS forecasts (Figure S4 in the supplementary material)." **(L: 414–419)**

We have also proofread the manuscript. Our replies to your comments and suggestions are as follows.

Specific comments

Line 62: I still think the fact that this regional model has about the same resolution as some of the global models should be explicitly mentioned. Perhaps changing "coupled models with lower computational cost" to something like "coupled models at a similar resolution, but with lower computational cost".

Response: Revised. **(L: 64)**

Line 99: Suggest adding "for vertical mixing" after "K-profile parameterization".

Response: Revised. **(L: 101)**

Lines 107-110: I don't understand what the authors mean by "a boundary layer between the ice shelf and ocean is formed". Are they referring to the oceanic boundary layer underneath the ice/ocean interface where the turbulent transfer is parameterized by the three equation formulation (Losch 2008, Holland and Jenkins 1999)? Also, since this paper is focusing more on sea ice outside the ice shelf cavities, is the sentence about partial cells necessary?

Response: We revised the sentence to "An oceanic boundary layer underneath the ice-shelf/ocean interface is formed following three physical constraints: the interface must be at the freezing point and both heat and salt must be conserved at the interface (Holland and Jenkins, 1999)". We deleted the sentence about partial cells. **(L: 110–111)**

Lines 126-129: What does the simulated ice shelf basal melt look like? I did not see it mentioned in Zhao et al. (2023).

Response: Zhao et al. (2023) did not provide any information on ice shelf basal melt. Figure 1 shows the modeled ice shelf basal melting rate during 2003–2020 derived from the data used in Zhao et al. (2023). The ice shelf basal melting rate is larger in the western Antarctic than that in the eastern Antarctic.

[Figure]

Figure 1. Ice shelf basal melting rate during 2003–2020 in the model free run.

Line 138: Do the authors have any references for studies that show LESTKF is suitable as mentioned here?

Response: We cited "(Vetra-Carvalho et al., 2018)". **(L: 141)**

Lines 162-163: I just wanted to thank the authors again for conducting the NoDA_Forecast and PE_Forecast runs.

Response: Thanks for your suggestion on these two runs. Your suggestions substantially improve the quality of this manuscript.

Line 250 and Figure 5: Why are the 24 hour and 168 hour NODA_Forecast IIEEs almost exactly the same?

Response: In the NoDA_Forecast run, the sea ice and ocean are strongly coupled, and the ice-ocean state evolution is uninterrupted, thus the difference between the 24h and 168h IIEEs is small. In the DA_Forecast run, we assimilate sea ice concentration observations to update the modeled sea ice concentration and thickness, but the top-layer ocean temperature is not updated. For example, we assume a case that the data assimilation step eliminates sea ice in a grid, but the top-layer ocean temperature of the grid is still close to freezing point, which leads to fast formation of sea ice in the following numerical integration steps. The data-assimilation-induced decoupled

sea ice and ocean states produce the obvious difference between the 24h and 168h IIEEs in the DA_Forecast run. Since we have already developed sea ice concentration-sea ice thickness-SST data assimilation scheme in the ArcIOPS system (Liang et al., 2019), the above-mentioned decoupled problem in the SOIPS will be solved in future work.

Line 295-297: Could the positive thickness bias in the western Ross Sea be a dynamic effect where the sea ice growth in that region is actually OK, but not enough ice is being advected out of that region? Figure 10 shows the magnitude bias in the Ross Sea sea ice drift is negative almost every month. Have the authors compared satellite-based estimates of the ice production and/or transport in the area (e.g. Drucker et al., 2011; Nihashi et al., 2017; Tian et al., 2020) with their estimates?

Response: We did not validate the modeled ice volume seasonal evolution due to the lack of satellite sea ice thickness observations with high spatial-temporal coverage in the Southern Ocean. In Zhao et al. (2023), we compared the modeled ice area seasonal evolution with the AMSR family data, and the result shows that the modeled sea ice area evolution is generally in line with the observations in the Ross Sea sector (Figure 2). In the model physics, sea ice dynamic equation determines that positive ice thickness bias is always accompanied with negative ice drift speed. The biases in ice thickness and ice drift speed forecasts could be reduced by assimilating sea ice thickness observations in future work.

We revised the sentence to "Satellite observations of sea ice concentration, thickness and drift have been used to estimate sea ice production and transport in the Antarctic coastal polynyas (Drucker et al., 2011; Nihashi et al., 2017; Tian et al., 2020). However, due to the relative scarce coverage of satellite observations in the Antarctic, especially in sea ice thickness, the evaluation of the SOIPS sea ice forecasts in this work still has considerable uncertainties". **(L: 438–441)**

[Figure]

Figure 2. The 2003–2020 mean annual cycle of the regional sea ice area in $10^6$ km$^2$. The solid and

dashed lines denote the observations and the model free run. The blue, orange, yellow, and purple lines denote the Amundsen-Bellingshausen Seas (ABS) sector, the Ross Sea (RS) sector, the Weddell Sea (WS) sector, and the Indian-Western Pacific Oceans (IwPO) sector, respectively. The observations are derived from the AMSR family sea ice concentration data between 2003 and 2020. (This figure is the same to Figure 3b in Zhao et al., 2023)

Lines 348-349: In January-March, the drift magnitude bias (Fig. 10) appears to be relatively small compared to the other months, not large.

Response: We revised the sentence to "During January–March, the Antarctic sea ice zone shrinks to its annual minima, sea ice drift magnitude bias appears to be relatively small compared to the other months". **(L: 353–354)**

Technical corrections

As before, this is not a complete list and there are many minor grammatical errors not listed that should be cleaned up in the next version.

Line 17: Suggest changing "the OSISAF data" to "non-assimilated OSISAF data".

Response: Revised.

Line 32: Suggest changing "safety maritime navigation" to "safe maritime navigation".

Response: Revised.

Line 123: "have been successfully" should be "has been successfully".

Response: Revised.

Line 214: Suggest changing "forecasts mainly locate" to "forecasts are mainly located"

Response: Revised.

Line 218: "that in some nearshore areas" should be "in some nearshore areas".

Response: Revised.

Line 254: Suggest changing "biases locate" to "biases are located".

Response: Revised.

Line 281: Suggest changing "changes relatively small in" to "changes a relatively small amount in".

Response: Revised.

Line 289: Suggest changing "locating" to "located".

Response: Revised.

Lines 334-335: Suggest changing "Along with the prolong of the forecast lead time" to "As the forecast lead time increases"

Response: Revised.

Line 344: Suggest adding "A" before "Previous study".

Response: Revised.

Line 358: Suggest changing "locates at" to "is located at".

Response: Revised.

Line 394: "find that the SOIPS" should be "finds that the SOIPS".

Response: Revised.

Line 397: Suggest changing "mainly locate" to "are mainly located".

Response: Revised.

Lines 421-423: I do not understand the point the authors are trying to make with this sentence. Is this related to the preceding sentence about needing to add landfast ice processes or the following sentence about static ice shelves?

Response: We deleted the sentence.

Line 430: I am not sure what the authors mean by "coastline sharp".

Response: Revised to "coastline".

Line 441: Should "dealing oceanic boundary" be "dealing with oceanic boundary"?

Response: Revised.

Anonymous Referee #2
Review comment for 'Southern Ocean Ice Prediction System version 1.0 (SOIPS v1.0): description of the system and evaluation of synoptic-scale sea ice forecasts' in Geoscientific Model Development
[Round 2]

The manuscript by Zhao et al improves significantly after some careful modification. Specifically, the authors compared the data assimilation (DA) experiment with the model free run (NoDA) to show the benefit of sea ice concentration DA in prediction. Also, comparison with the persistence forecast highlights the advantages of dynamical prediction, especially when the sea ice changes quickly. The authors also dig into the model's systematic error and use it to explain the model's deficiency in the onset-to-fast freezing period. I appreciate the author's effort in addressing my concern. I suggest the manuscript be published after some extra modifications.

Response:
Dear reviewer, thanks a lot for your time and valuable comments on this manuscript. We have revised the manuscript according to your precious comments. Our replies to your comments and suggestions are as follows.

Major comment:
1. As the revised manuscript includes some more content, e.g., a comparison with the model free run and the persistence forecast, the abstract needs to be updated accordingly.

Response: We added the statement of "The comparison among the persistence forecasts, the SOIPS forecasts with and without data assimilation further shows that both model physics and data assimilation scheme play important roles in reliable sea ice forecasts in the Southern Ocean." into the abstract. **(L: 23–25)**

2. According to Figure 9, the sea ice drift prediction improves a lot due to the assimilation of sea ice concentration. Could you explain the reason for it? Is this related to the improved sea ice thickness, the sea ice concentration, or both (Chapter 6.1.1 in Leppäranta M. The drift of sea ice[M]. Springer Science & Business Media, 2011)?

Response: We added the statement of "The improvement of sea ice drift forecasts originates in principle from the enhancement of the SOIPS forecasts on sea ice concentration and thickness induced by data assimilation of the observed sea ice concentration, since sea ice drift is impacted by both sea ice concentration and thickness (Leppäranta, 2011)." into the main text. **(L: 336–339)**

3. For Section 3.5, a specific case is shown to illustrate the skill of the system to predict the sea ice convergence and divergence. Please elaborate more on the source

of this skill. For example, does this skill appear in the NoDA experiment? If so, we can suspect that this skill originates largely from the atmosphere forcing.

Response: Figure 1 shows that this skill originates largely from the atmosphere forcing. We added the statement of "Further analysis shows that the forecasting skill of sea ice convergence originates largely from the precise atmosphere forcing rather than the effects of sea ice concentration data assimilation (not shown)" into the main text. **(L: 388–390)**

[Figure]

Figure 1. Sea ice convergence rate in the NoDA_Forecast (left column) and DA_Forecast (right column) runs. The top, middle, and bottom panels denote forecasts on November 19, 20, and 21, 2021, respectively. The forecasts were initialized on November 18, 2021. Black arrows denote sea ice drift vectors, while red and blue contours indicate that sea ice drift in the corresponding area tends to convergent and divergent, respectively. The red dot in each figure marks the Antarctic

China Zhongshan Station.

Minor comment:
Line 102, sever as.

Response: Revised.

Line 175, please elaborate on the purpose of comparing the SOIPS forecasts to the physical analysis field of the Antarctic Ocean produced by MOI. I didn't get the idea of this comparison.

Response: Another reviewer suggested that some intercomparisons between the SOIPS system and other ice forecasts systems are appreciated in the revised manuscript. We can not download operational sea ice forecasting products from the MOI and GIOPS. The MOI physical analysis field of the Antarctic Ocean can be accessed freely on the CMEMS website.

Line 265, by definition, the IIEE is the value of the error. The contour in Figure 6 should be the mismatch between these two data.

Response: Revised.